# λ/30 inorganic features achieved by multi-photon 3D lithography

Feng Jin [1✉], Jie Liu[1], Yuan-Yuan Zhao[2], Xian-Zi Dong[1], Mei-Ling Zheng [1,3✉] & Xuan-Ming Duan [2✉]

It's critically important to construct arbitrary inorganic features with high resolution. As an inorganic photoresist, hydrogen silsesquioxane (HSQ) has been patterned by irradiation sources with short wavelength, such as EUV and electron beam. However, the fabrication of three- dimensional nanoscale HSQ features utilizing infrared light sources is still challenging. Here, we demonstrate femtosecond laser direct writing (FsLDW) of HSQ through multi-photon absorption process. 26 nm feature size is achieved by using 780 nm fs laser, indicating super-diffraction limit photolithography of λ/30 for HSQ. HSQ microstructures by FsLDW possess nanoscale resolution, smooth surface, and thermal stability up to 600 °C. Furthermore, we perform FsLDW of HSQ to construct structural colour and Fresnel lens with desirable optical properties, thermal and chemical resistance. This study demonstrates that inorganic features can be flexibly achieved by FsLDW of HSQ, which would be prospective for fabricating micro-nano devices requiring nanoscale resolution, thermal and chemical resistance.

[1] Laboratory of Organic NanoPhotonics and CAS Key Laboratory of Bio-inspired Materials and Interfacial Science, Technical Institute of Physics and Chemistry, Chinese Academy of Sciences, No. 29, Zhongguancun East Road, Haidian District, Beijing 100190, P. R. China. [2] Guangdong Provincial Key Laboratory of Optical Fiber Sensing and Communications, Institute of Photonics Technology, Jinan University, 855 East Xingye Avenue, Panyu District, Guangzhou 511443, P. R. China. [3] School of Future Technologies, University of Chinese Academy of Sciences, Yanqihu Campus, Huairou District, Beijing 101407, P. R. China. ✉email: jinfeng@mail.ipc.ac.cn; zhengmeiling@mail.ipc.ac.cn; xmduan@jnu.edu.cn

Three-dimensional (3D) inorganic features have attracted considerable attention in recent years for the wide applications in thermal management, architected materials, solar cells, quantum chips, and aeronautics[1–5]. However, the manufacture of 3D inorganic features is hindered by the limitation of intrinsic crystallization of the precursors, which results in rare report on directly constructing of 3D inorganic features[6]. Through pyrolysis of 3D features constructed by 3D printing of photoresists containing inorganic precursors and/or nanoparticles, various 3D inorganic features have been successfully achieved[7–11]. Nonetheless, the feature size of inorganic microstructures obtained by 3D printing technology has been generally limited to a few micrometers, which is unsuitable for the applications requiring nanoscale resolution[12,13]. Therefore, it's important to construct 3D inorganic features with nanoscale resolution.

Femtosecond laser direct writing (FsLDW) is a versatile 3D micro and nano-printing tool for the fabrication of 3D fine microstructures. By employing multi-photon absorption process, arbitrary 3D fine features at a resolution of a few tens of nanometers can be readily achieved by FsLDW[14–20]. Under femtosecond (fs) laser irradiation, photocleavage of photoactive molecules and subsequent polymerization or cleavage will induce different solubility of the photoresists, resulting in the fabrication of well-defined microstructures[21–24]. Especially, with very high laser intensity, fs laser pulse is capable of directly breaking the chemical bond of monomer molecules and triggering the crosslinking of organic photoresists without any photoinitiators via multi-photon ionization (MPI) process[25,26]. Analogous to the organic photoresist, hybrid organic/inorganic photoresist SZ2080 has been fabricated by employing FsLDW without the addition of photoinitiator[27,28]. Recently, R. G. Hobbs et al. has demonstrated the nanostructuring of inorganic photoresist hydrogen silsesquioxane (HSQ) by hot-electron emission, which was induced by fs laser exposure of gold nanorods[29]. As an important inorganic e-beam resist, HSQ possesses the advantages of sub-10 nm resolution, optical transparency, electronic insulation, and low dielectric constant[30–34]. Moreover, exposed HSQ exhibits similar property to silicon oxide, facilitating its potential applications in nanophotonic and nanoelectronic devices[35,36]. However, the fabrication of 3D HSQ features via FsLDW without any assistance has not been reported, which is mainly attributed to the inherent absorption of HSQ locating at the deep UV region. It has been demonstrated that HSQ did not show measureable sensitivity for wavelength range between 800 nm and 193 nm, but only acted as negative tone photoresist under radiation sources with short wavelengths, such as DUV light at 157 nm, EUV light at 13.5 nm, proton beam, ion-beam, X-ray and electron beam[37]. Although photoinitiators could be employed to endow HSQ with photosensitivity as the previously reported inorganic photoresist[38], residual photoinitiators in the polymerized features would cause unfavourable results, including either autofluorescence needed to be eliminated[39], or pollution problems in the construction of

semiconductor devices[40]. In addition, although 3D HSQ features are important, it's still challenging to achieve 3D HSQ features, which generally needs complex strategy and time-consuming process[41–44]. Thus, the following question arises: can we induce nanostructuring of HSQ by using NIR fs laser pulse without employing foreign specimen, especially for the fabrication of 3D features with nanoscale feature size? Furthermore, the mechanism for FsLDW of inorganic HSQ photoresist has to be figured out.

Here, we demonstrate FsLDW of HSQ for 3D nanoscale features by using 780 nm fs laser via multi-photon absorption process. HSQ feature with 26 nm feature size, about $\lambda/30$ of the wavelength of irradiation source, has been successfully achieved, indicating the FsLDW fabrication capability beyond the optical diffraction limit. Raman microscopy and Fourier transform infrared (FT-IR) spectroscopy were used to explore the mechanism of FsLDW of HSQ, which is attributed to MPI induced photocleavage of Si-H bonds in HSQ and subsequent crosslinking of HSQ. 2D and 3D HSQ microstructures were fabricated to depict the versatility of FsLDW in fabricating arbitrary HSQ microstructures with nanoscale resolution and excellent surface roughness. The 3D HSQ microstructure by FsLDW exhibited excellent thermal resistance up to 600 °C. Furthermore, we perform FsLDW to construct biomimetic structural colour, Fresnel lenses and diffractive gratings, which exhibit excellent optical property, thermal stability, and chemical resistance. This work would widen the choice of materials for FsLDW, enabling the creation of 3D inorganic microstructures with nanoscale feature size for harsh environments.

## Results

**Femtosecond laser direct writing of inorganic photoresist HSQ**. We conducted FsLDW of HSQ by employing a 780 nm fs laser equipped on a home-built apparatus described in our previous work (Methods, Supplementary Fig. 1)[45]. Figure 1 is the schematic of the FsLDW of HSQ microstructures. When the 780 nm fs laser pulse is irradiated on the HSQ film obtained by spin-coating and baking, nonlinear absorption would occur since HSQ does not absorb light in the wavelength range between 200 and 800 nm (Supplementary Fig. 2), instead exhibits photocuring sensitivity at 157 nm[37]. The nonliear absorption process is further verified by the in-site measurement of the nonlinear absorption order in HSQ[46–48], indicating multi-photon absorption in HSQ by FsLDW (Supplementary Fig. 3). Although no photoinitiator exists in HSQ, photocuring of HSQ is achieved by multi-photon absorption triggered cleavage of Si-H bonds and subsequent crosslinking of HSQ, resulting in the fabrication of 3D nanoscale HSQ features.

We demonstrate the FsLDW of HSQ with nanoscale feature size by employing threshold effect. It's well known that threshold effect exists in the nonlinear absorption initiated photoreaction[14,25,49]. The threshold effect possesses highly nonlinear dependence on fs laser intensity when the laser intensity is approaching to the threshold. Effective intensity profile becomes

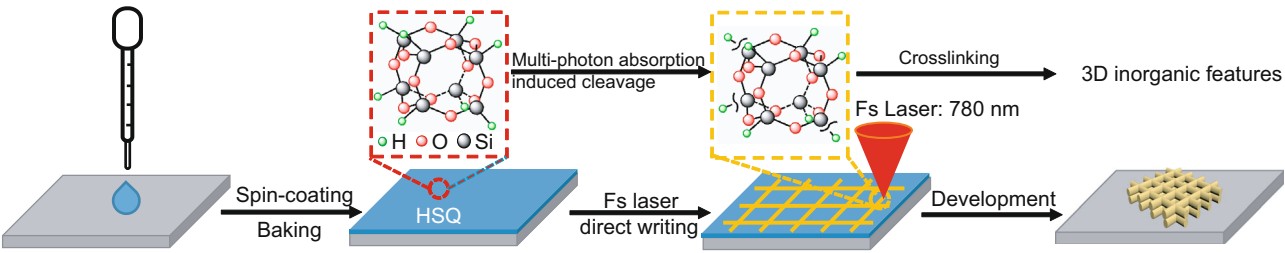

**Fig. 1 Multi-photon 3D lithography of inorganic features.** Schematic of the fabrication of 3D nanoscale inorganic features by femtosecond laser direct writing of HSQ.

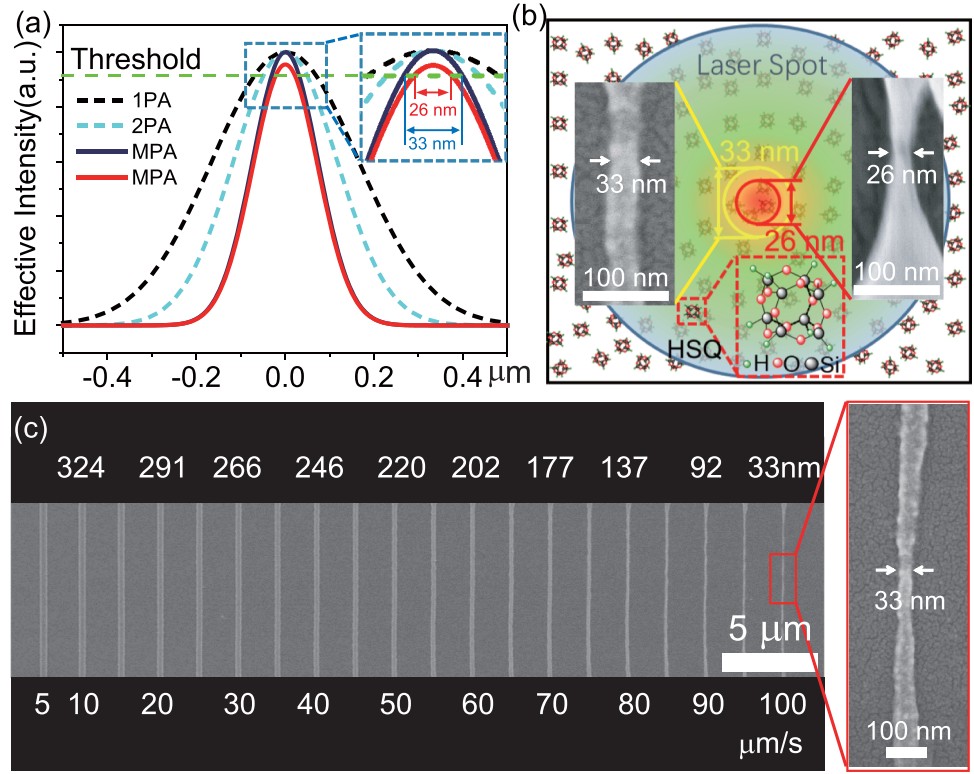

**Fig. 2 The construction of nanoscale HSQ features by FsLDW. a** Illustration of patterning HSQ with sub-diffraction feature size by FsLDW through nonlinear absorption process. The dashed green line denotes the threshold value of the fs laser induced polymerization. **b** Scheme of HSQ features fabricated by FsLDW with single-scanning method and cross-scanning method, resulting in 33 nm and 26 nm feature sizes, respectively. When the laser intensity increases, only a portion of the laser spot exceeds the threshold for polymerization, enabling the creation of sub-diffraction features. The blue circle represents the fs laser focus spot size. The inset is the molecular structure of HSQ. **c** SEM image of HSQ nanowires fabricated by FsLDW with different laser scanning speeds. The inset is the HSQ nanowire with the narrowest linewidth of 33 nm.

narrower and steeper by increasing the number of photon simultaneously absorbed (Fig. 2a). The laser intensity could be precisely controlled so that only a small portion of the focus spot exceeds the threshold for polymerization, resulting in feature size beyond optical diffraction limit. Through precisely tuning the laser intensity and scanning speed, 33 nm HSQ nanowire has been fabricated by FsLDW via single-scanning method (Fig. 2b, c). AFM image depicts the height of 30 nm for the HSQ nanowire fabricated by the scanning speed of 100 μm/s (Supplementary Fig. 4). The construction of freelying HSQ nanowire with the width of 33 nm and the height of 30 nm verifies the beyond diffraction-limit fabrication of HSQ by FsLDW. In order to persue the limit of feature size of HSQ by FsLDW, cross-scanning method was employed to further decrease the feature size. Freelying HSQ feature on the substrate with the feature size of 26 nm has been successfully constructed (Fig. 2b, Supplementary Figs. 5b, 6f, 7), which is only λ/30 of the 780 nm fs laser. The 26 nm HSQ feature has sharp boundary and smooth edge, indicating that even smaller feature size could be achieved by approaching closer to laser intensity threshold. It's worth noting that 33 nm and 26 nm HSQ features are both freelying microstructures, which are directly constructed by FsLDW on the substrate. To the best of our knowledge, it is the first time that HSQ has been patterned by using NIR fs laser without any assistance.

We further depict the formation mechanism of 33 nm and 26 nm HSQ features by FsLDW. For the FsLDW of HSQ by single-scanning method with 2 μm spacing (Supplementary Fig. 5a), the interference of adjacent nanowires is negligible since the spacing between adjacent lines is much larger than λ/N.A.

(about 538 nm). In this case, crosslinking of HSQ occurs at the central region of the focus spot, while HSQ oligomers appear at the fringe of the focus spot due to below-threshold irradiation. Instead of the fringe, the central region of the focus spot will produce HSQ features after development, resulting in the formation of freelying sub-diffraction 33 nm HSQ feature due to the threshold effect[14,25,49]. Nonetheless, if we conduct the FsLDW of HSQ by cross-scanning method with 0.5 μm spacing (Supplementary Fig. 5b), the interference of adjacent nanowires occurs since the spacing between adjacent lines is similar to λ/N.A. When the fs laser focus spot scans in the perpendicular direction, the crosslinking degree of HSQ in the laser exposed region can be enhanced by the existed HSQ oligomers caused by the fringe of the focus spot in the first laser scanning. HSQ features with adequate crosslinking degree can be constructed by reducing the laser exposure closer to the laser intensity threshold (Fig. 2a), resulting in the formation of freelying 26 nm HSQ feature on the substrate. As a result, λ/30 feature size of HSQ by FsLDW is successfully achieved, which is significant in theoretical research and potential applications for nanostructuring of inorganic features. Meanwhile, the size of the HSQ features could be greatly influenced by the experimental parameters, such as laser intensity fluctuation, focusing condition, roughness of the substrate, post-polymerization shrinkage, environmental humidity and storage time of HSQ. The controllable lithography of 26 nm HSQ features could be achieved by optimizing the experimental parameters.

The exposure mechanism of HSQ by FsLDW is investigated by employing Raman and FT-IR microscopy. Raman and FT-IR spectra were collected on unexposed and exposed regions of HSQ

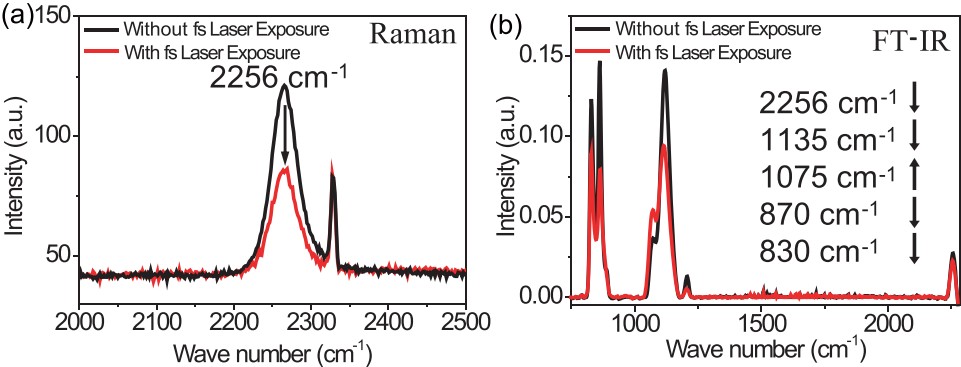

**Fig. 3 Femtosecond (fs) laser exposure induced change in Raman and FT-IR spectra of HSQ. a** Raman spectra of HSQ film before (black) and after (red) fs laser exposure. **b** FT-IR spectra of HSQ film before (black) and after (red) fs laser exposure.

film prior to development (Fig. 3). The decrease of the peaks centered at $2256\,cm^{-1}$ on Raman and FT-IR spectra identifies the photocleavage of Si-H bond of HSQ induced by fs laser exposure, which is similar to e-beam exposure and thermal treatment[50,51]. The drop of the bands around 830 and $870\,cm^{-1}$ on FT-IR spectra also confirms the reduction of Si-H content in the exposed HSQ film. Meanwhile, the shrinkage of band located at $1135\,cm^{-1}$, and the increase of the band at $1075\,cm^{-1}$ can be observed on the FT-IR spectra, which is indicative of the transformation from caged structure of the unexposed HSQ to the crosslinked network of the exposed region[33,50].

According to the FT-IR, Raman and UV-Vis absorption spectra, we suggest that the FsLDW of HSQ is mainly attributed to MPI induced by nonlinear absorption process. MPI would induce chemical bond breaking in HSQ when the laser intensity is above 1.4 TW/cm$^2$ (Supplementary Fig. 8), which is high enough to trigger photocleavage of chemical bonds according to previous reports[52–55]. Since the bond energy of Si-H (354.7 kJ/mol) is much weaker than that of Si-O-Si (422.5 kJ/mol)[56,57], it is expected that Si-H bonds in HSQ are readily to be broken via MPI, which is verified by the Raman and FT-IR spectra. The breaking of Si-H bonds further triggers the transfer of HSQ from cage to network configuration by the formation of Si-O-Si bond between adjacent HSQ cage molecules. The network configuration of HSQ will be undissolvable in the developer and left after development, while the cage configuration of HSQ will be removed by resolving in the developer. The phenomenon is contrast to the fs laser-induced ablation of silica, which is mainly caused by the breaking of Si-O-Si bonds induced by nonlinear absorption of high energy fs laser irradiation[58,59]. The key for the FsLDW of HSQ is to precisely control the laser intensity to stimulate localized polymerization instead of ablation. As a result, the inherent characteristic of ultra-fast and ultra-strong property endows NIR fs laser the capability of photocuring of HSQ, which permits us to get rid of the restriction of plasmonic substrate and photoinitiators[29,38].

Numerous 2D and 3D microstructures were fabricated to demonstrate the feasibility of constructing arbitary features of HSQ by FsLDW. Figure 4 presents SEM and AFM images of the fabricated 2D and 3D HSQ microstructures. In the nanowire array shown in Fig. 4a, parallel nanowires with the linewidth of 81 nm and the periodicity of 400 nm are achieved. The periodicity of the nanowires is as small as 400 nm, suitable for fabricating high-performance optical devices, for example, anti-reflective grating, and X-ray grating[60]. Nanodot array is fabricated with the dot diameters from 312 to 99 nm and the periodicity of 1 μm (Fig. 4b). In Fig. 4c, archimedes spiral array is fabricated, in which the linewidth monotonically decreases from the center to the outside due to the gradual increase of the scanning speed. The

narrowest linewidth of the archimedes spiral is about 125 nm. The letter NANO array is also fabricated with uniform linewidth, and the narrowest line of the character is 118 nm, as shown in Fig. 4d. In addition, 2D HSQ micro-square with superior morphology was fabricated by optimizing the fabrication conditions (Fig. 4e). AFM measurement of the microstructure shows very smooth surface with a roughness of about 3 nm (Fig. 4f). The surface roughness of HSQ by FsLDW is comparable with that of the HSQ microstructures fabricated by e-beam lithography, and that of the sintered glass microstructures by stereolithography, as well[7,61]. The successful fabrication of complex 2D features indicates that FsLDW is suitable for the construction of arbitary 2D HSQ features with high resolution and high smoothness, which is critical for optical and optoelectronic applications.

In addition to 2D features, FsLDW is also employed to fabricate 3D HSQ features, which is a distinctive characteristic of FsLDW. As shown in Fig. 4g–i, the double-layered grid microstructure is constructed by the periodic grids with the feature size from 65 to 98 nm. The bottom layer is stacked on the substrate, while the upper layer overlaps on the bottom layer by anchoring points and suspended features. Magnified SEM images and statistical analysis (Supplementary Fig. 9) indicate that the feature size of the upper layer is smaller than that of the bottom layer, which is probably due to the shrinkage of the HSQ features in the development process and surface proximity effects induced by the non-crosslinked HSQ oligomers. Furthermore, 3D woodpile microstructure has been fabricated by using FOX-16, which can make thicker HSQ film (Supplementary Fig. 10). The feasibility to construct 3D HSQ microstructures with nanoscale resolution is significant to fabricate nano-optics devices such as photonic crystals. Although 3D HSQ microstructures can be obtained by e-beam lithography, or ion-beam lithography[41–44], FsLDW provides a simple, robust method to fabricate arbitary 3D HSQ microstructures.

**Thermal stability of 3D HSQ microstructure fabricated by FsLDW.** The thermal stability of 3D HSQ microstructures fabricated by FsLDW was further investigated. Double layered grid microstructure (20 μm × 20 μm) was firstly baked on a hotplate at 400 °C in air for 0.5 h, and then heated in a tube oven in Ar atmosphere for 2 h at the temperature of 500, 600, and 700 °C, respectively. The 3D microstructure almost keeps the original shape and morphology without serious collapse and disintegration, although slight shrinkage and local deformation occur after heating to 600 °C (Fig. 5). The diameter of the small hole decreases from 149 nm to 138, 126, and 119 nm, respectively (Fig. 5a-a‴). SEM images in Fig. 5b-b‴ show that the periodicity

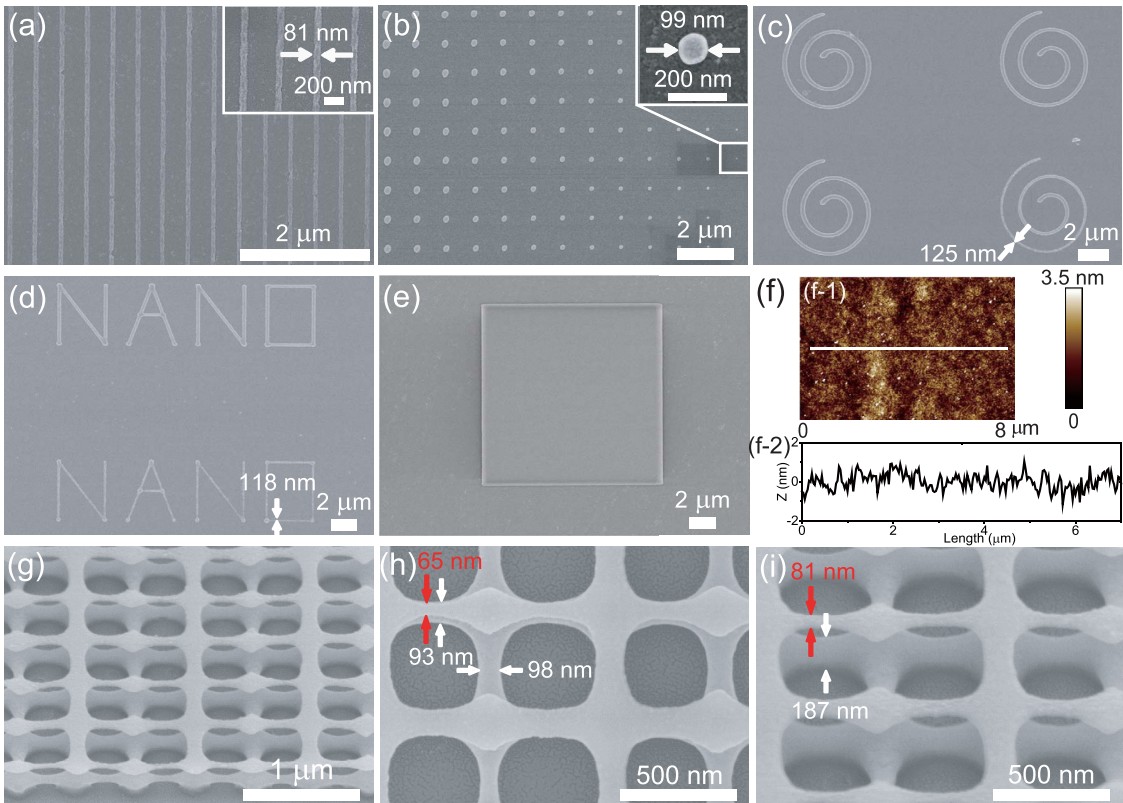

**Fig. 4 2D and 3D HSQ features fabricated by FsLDW. a** SEM image of nanowire array. The inset is the magnified SEM image of the nanowire array. **b** SEM image of nanodot array. The inset is the magnified SEM image of the nanodot. **c** SEM image of mircro-archimedes spiral array. **d** SEM image of micro-letters of NANO array. **e** SEM image of micro-square. **f** AFM measurement on the HSQ micro-square shown in **e**. **f-1** AFM image of the micro-square. **f-2** The profile of the top surface of the HSQ micro-square shown in **f-1**, exhibiting the roughness of about 3 nm. **g** 45°-tilted-view SEM image of the 3D double layered HSQ microstructure. Magnified **h** top view and **i** 45°-tilted-view SEM images of the 3D HSQ microstructure. The feature sizes of the upper layer (red) and bottom layer (white) are shown in **h** and **i**.

in horizontal direction decreases from 1008 nm (pristine), to 1005 nm (400 °C), 1001 nm (500 °C), and 940 nm (600 °C), respectively. Meanwhile, the height of the microstructure drops from 439 nm to 386, 367, and 350 nm, respectively (Fig. 5d-d'''). The shrinkage of the 3D microstructure is listed in Supplementary Table 1. Obviously, the change of the shape and morphology of 3D HSQ microstructure is negligible when it is heated to 400 °C in air. Very tiny holes appear on the surface of the 3D HSQ microstructure after being baked at 400 °C in air, which is probably attributed to the output of gas during thermal treatment[51,62]. The appearance of tiny holes and the negligible change in shape in the 3D HSQ microstructure indicate the transfer of HSQ to silicon oxide[51]. Further increase the temperature to 500 and 600 °C, the morphology change of the 3D HSQ microstructure is gradually nonnegligible, with the slight disintegration marked by the red circles (Fig. 5a'', a''', d'', and d'''). Nevertheless, we did not observe the breaking of the features in the 3D HSQ microstructure even for the thinnest freestanding and suspended features up to 600 °C (Fig. 5, Supplementary Fig. 11a–l). Further increase the temperature to 700 °C, only partial 3D HSQ microstructure can be recognized (Supplementary Fig. 11m–o) due to the distortion of the commerical glass substrate. By contrast, thermal unstability caused by thermal decomposition at 450 °C has been observed for traditional organic and inorganic/organic hybrid photoresists, and would hinder the practical applications in high-temperature devices[63,64]. As a result, HSQ is superior to organic photoresists in thermal performance. The thermal performance of HSQ microstructure by FsLDW is suitable for directly constructing high-temperature

nano-optics and nano-electronic devices, and fabricating templates requiring thermal stability, as well.

**Thermal and chemical resistance of HSQ optical microstructures by FsLDW.** Biomimetic structural colour with thermal resistance is achieved via HSQ 3D microstructure fabricated by FsLDW. Structural colour is frequently observed in some butterfly swings[65], in which alternative layers of compact layers and air are periodically stacked, yielding interference effects that cause structural coloration. Inspired by the butterfly swings, HSQ double-layer grid microstructure is designed and fabricated by FsLDW to realize structural colour, as schemed in Fig. 6a. The periodicity of the double-layer grid in horizontal direction is 1 μm. The morphology of the 3D HSQ microstructure is shown in Fig. 6b, in which double-layer grids are well constructed. The upper layer is stacked on the bottom layer, forming the HSQ-air-HSQ structure. AFM image depicts the uniform periodic microstructure with the thickness of 500 nm (Supplementary Fig. 12). Due to the multi-layer interference of the periodic HSQ microstructures, Bragg peak emerges on the reflection spectrum with the central wavelength of 631 nm (Fig. 6c). Structural colour of orange is observed from the microstructure by employing an optical microscope under reflection mode (inset of Fig. 6c).

Furthermore, the 3D HSQ grid microstructure is heated at 600 °C in Ar atmosphere for 1 h to evaluate the thermal stability of the structural coloration. Thermal treatment did not eliminate the structural colour of the 3D HSQ grid microstructure, but resulted in a blue-shift of the structural colour from orange to

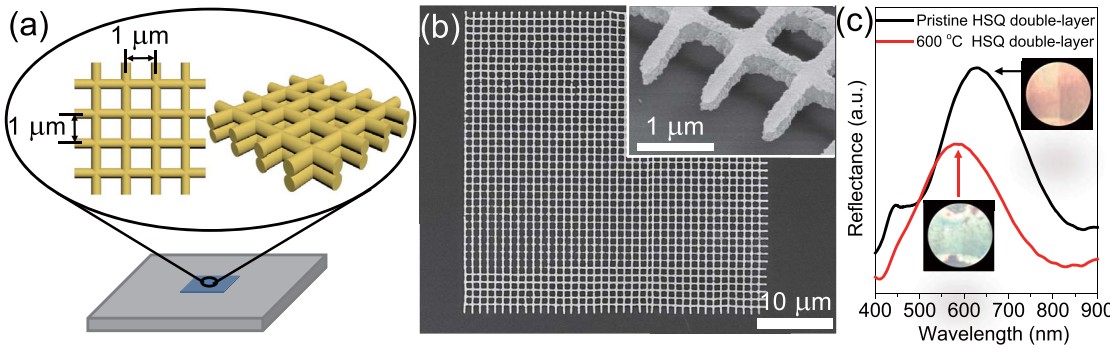

**Fig. 5 SEM images of double layered grid HSQ microstructure by FsLDW before and after thermal treatment. a–d** are SEM images of the HSQ microstructure before thermal treatment. **a′–d′**, **a″–d″**, and **a‴–d‴** are SEM images of the HSQ microstructure after thermal treatment at 400, 500, and 600 °C, respectively. The insets in **a–a‴**, **c–c‴** are the high-magnification details. The red dashed circles point out the slight decomposition of the 3D HSQ microstructure.

**Fig. 6 Biomimetic structural colour with thermal resistance via HSQ double-layer microstructure constructed by FsLDW. a** Schematic illustration of the configuration design of the HSQ double-layer microstructure with structural colour. **b** SEM image of HSQ double-layer microstructure constructed by FsLDW. The inset is the magnified tilted SEM image of the HSQ double-layer microstructure. **c** Reflectance spectra of the HSQ double-layer microstructure before (black) and after (red) 600 °C treatment. The inset exhibit the structural colour of the double-layer HSQ microstructure before and after 600 °C treatment.

yellow green (inset of Fig. 6c). Accordingly, the Bragg peak is transferred from 631 to 583 nm (Fig. 6c). Since the refractive index is slightly decreased from 1.380 (multi-photon cured HSQ, Supplementary Fig. S13) to 1.375 (thermal treatment of multi-photon curved HSQ at 600 °C, Supplementary Fig. S13), the blue-shift of the Bragg peak is mainly attributed to the decreased refractive index and the shrinkage of the HSQ double-layer

microstructure during the heating process. Although structural colour has been achieved by the combination of metal layers and HSQ microstructures constructed by e-beam lithography[30,66], HSQ microstructures by FsLDW provide a simple and direct method to construct structural colour. The biomimetic structural colour of HSQ microstructures with thermal resistance will exhibit potential widespread applications in harsh conditions.

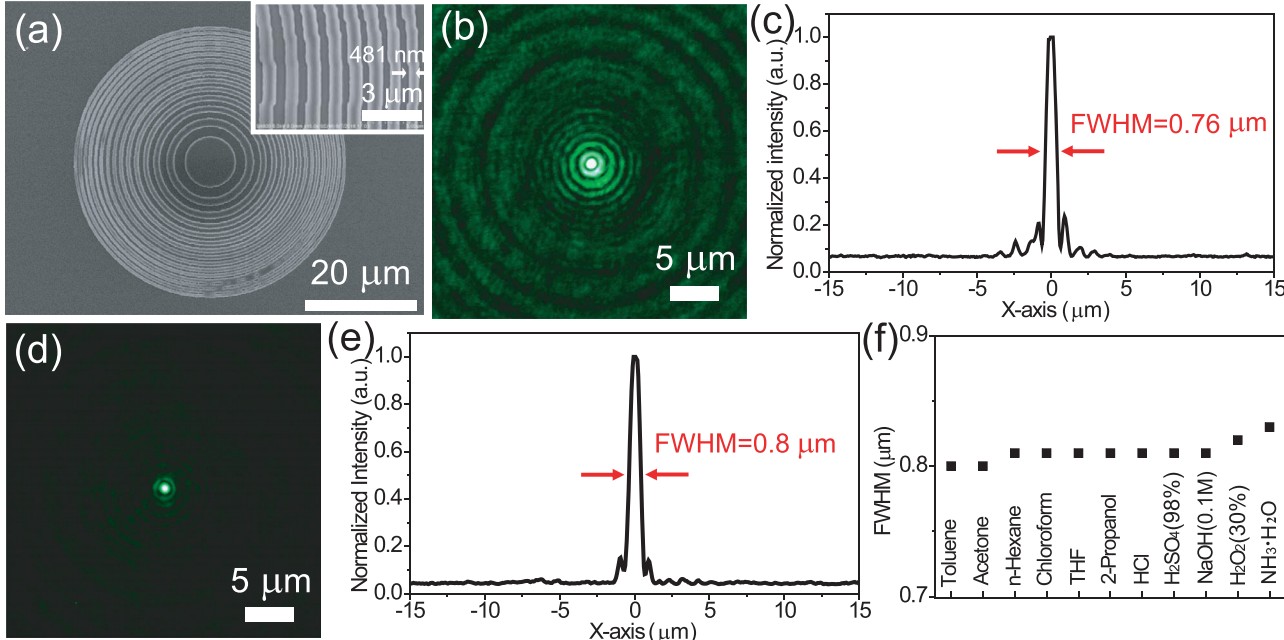

**Fig. 7 Thermal and chemical resistance of HSQ Fresnel lens by FsLDW. a** SEM image of the HSQ Fresnel lens microstructure by FsLDW. The inset is the magnified SEM image. **b** The microscope focusing image of the HSQ Fresnel lens. **c** The intensity distribution collected along the *x*-axis direction shown in **b**. **d** The microscopic focusing image of the HSQ Fresnel lens after heating at 400 °C for 0.5 h. **e** The intensity distribution collected along the *x*-axis direction shown in **d**. **f** Full-width at half maximum (FWHM) of the optical intensity profile along *x*-axis of the focusing spot of HSQ Fresnel lens after being exposed to different chemical reagents, which are selected to imitate harsh conditions.

The potential application of HSQ microstructures by FsLDW is further demonstrated by the construction of prototype optical devices of Fresnel lens and diffractive gratings (Fig. 7 and Supplementary Fig. 17). The HSQ Fresnel lens is designed and constructed by 15 concentric rings (Fig. 7a). The concentric rings are smooth and isolated, and the smallest linewidth of the rings and the narrowest space between the rings are 481 and 350 nm, respectively. Meanwhile, the radius of the central zone and the whole Fresnel lens is determined to be 4.84 and 25.4 μm, respectively. The focal length of the Fresnel lens can be calculated by the equation: $f = r^2/m\lambda$, where *r* is the radius of the central zone, *λ* is the wavelength of the incident light, and *m* is an integer value which indicates the zone number. Hence, the focal length of the Fresnel lens is calculated to be 44 μm at the wavelength of 532 nm. The numerical aperture (N.A.) of the Fresnel lens is 0.49. We measured the focusing property of the Fresnel lens by employing a home-built experimental setup (Supplementary Fig. 14). Figure 7b shows the focusing image of the Fresnel lens, in which bright focusing spot emerges with symmetrically circular shape. Figure 7c is the optical intensity profile of the far field pattern of the focusing spot along x-axis. The full-width at half maximum (FWHM) of optical intensity profile for the fabricated Fresnel lens is 0.76 μm (0.7 *λ*/N.A.), indicating good focusing characteristic of the HSQ Fresnel lens.

We demonstrate the thermal and chemical resistance of the HSQ Fresnel lens by evaluating the focusing feature after thermal and chemical treatment. Firstly, we measured the thermal resistance of the HSQ Fresnel lens by characterizing the focusing feature after heating it on a hotplate at 400 °C in air for 0.5 h. The far field pattern of the focusing spot suggests that the focusing property is slightly affected by the thermal treatment (Fig. 7d, e). The FWHM of optical intensity profile along x-axis is 0.8 μm (0.74 *λ*/N.A.) after thermal treatment. The 5% increase of FWHM indicates good thermal resistance of the HSQ Fresnel lens. The slight deterioration of the focusing feature is ascribed to the slight break of the HSQ Fresnel lens under high temperature treatment

in air (Supplementary Fig. 15). The slight break of the HSQ Fresnel lens is probably caused by the residual internal stress during the polymerization of the HSQ, and outgas of the HSQ at the temperature of 400 °C[50,61].

After thermal treatment at 400 °C, we further measured chemical resistance of the HSQ Fresnel lens by evaluating the focusing property of the HSQ Fresnel lens after being exposed to chemicals. Figure 7f and Supplementary Fig. 16 depict the focusing feature of the HSQ Fresnel lens after being soaked in each representative chemical reagent for 1 h. Negligible change in focusing feature has been observed for the HSQ Fresnel lens before and after chemical treatment, indicating that obvious swelling, degradation or optical alteration did not occur in the HSQ Fresnel lens even for 98% $H_2SO_4$. The chemical resistance of the heated HSQ Fresnel lens is mainly attributed to the transfer of the microstructure from crosslinked HSQ to silicon oxide during the thermal treatment at 400 °C[51,61]. The good thermal and chemical resistance of HSQ microstructures facilitate the wide applications of HSQ in the integration with traditional optical devices, especially for the usage in harsh environments.

## Discussion

We demonstrate 780 nm fs laser direct writing of nanoscale inorganic features by using inorganic photoresist HSQ, which is triggered by multi-photon absorption process. The feature size of HSQ by FsLDW is 26 nm, which is about *λ*/30 of the exposure source and breaks the optical diffraction limit. The exposure mechanism of HSQ by FsLDW is attributed to the photocleavage of Si-H bond and subsequent crosslinking of HSQ by the formation of Si-O-Si bond through fs laser induced multi-photon ionization. 2D and 3D features were fabricated by FsLDW to exhibit the feasibility of constructing arbitrary HSQ microstructures with high resolution and smooth surface. HSQ microstructures exhibited superior thermal stability up to 600 °C. Furthermore, FsLDW is employed to construct biomimetic structural colour, Fresnel lens and diffractive gratings, which

exhibit excellent optical properties, thermal and chemical resistance. The FsLDW of HSQ broadly expands the excitation source for the patterning of HSQ, which would be prospective for fabricating micro-nano devices for harsh circumstances.

## Methods

**Materials**. HSQ (XR-1541-004) and FOX-16 were purchased from Dow Corning Company (USA) and kept at 4 °C in a refrigerator. Aqueous solution of tetramethylammonium hydroxide (TMAH, 0.26 N) was purchased from Shipley Company (USA). Glass substrates (24 mm × 40 mm × 0.15 mm) were purchased from Yanchengfeizhou Company. Silicon wafers (4 inch) and quartz substrates (10 mm × 10 mm × 1 mm) were purchased from Beijing Zhongjingkeyi Technology Co., Ltd. Analytical reagents of toluene, chloroform, n-hexane, acetone, tetrahydrofuran, 2-propanol, sodium hydroxide, hydrochloric acid (HCl, 37%), sulphuric acid ($H_2SO_4$, 98%), ammonia solution (25%), and hydrogen peroxide ($H_2O_2$, 30% aqueous solution) were purchased from Sinopharm Chemical Reagent Co. Ltd and used as received without further purification.

Silicon wafers, glass and quartz substrates were treated with Piranha solution ($H_2SO_4$/30% $H_2O_2$ (v/v) = 7/3) to remove the organic residues. Caution: piranha solution is a strong oxidant and should be handled with care. After treatment, the substrates were washed and sonicated in ultrapure water (18 MΩ·cm, Mini Q) for several times, and dried by $N_2$. HSQ films were deposited on the substrates by spin-coating of HSQ at 4000 rpm for 1 min on a spin coater (KW-4A, Institute of Microelectronics of the Chinese Academy of Sciences). For the fabrication of 3D HSQ microstructures, the HSQ films were spin-coated at 1000 rpm for 1 min. The HSQ films were prebaked on the hotplates (CT-946, Beijing Zhongjingkeyi Technology Co., Ltd) for 5 min at 60 °C, and another 3 min at 90 °C to remove the solvent. The temperature of 90 °C is selected for prebake since higher temperature would induce the transfer of HSQ from caged form to crosslinked form.

**Femtosecond laser direct writing of HSQ**. FsLDW of HSQ is performed by employing a 780 nm fs laser (120 fs, 80 MHz) equipped on a home-built apparatus described in a previous report[45]. The optical setup is shown in Supplementary Fig. 1a, in which the 780 nm fs laser beam is focused by an oil-immersion objective lens (N.A. = 1.45, 100 ×, Olympus). By accurately tuning the laser intensity, scanning speed, and exposure position, the laser exposure conditions of local region in HSQ film can be precisely controlled. After laser exposure, the HSQ film was developed in the developer (0.26 N TMAH) for 3 min, and then ultrapure water for 1 min. The residual water was removed by blowing the sample with $N_2$ to obtain the fabricated microstructures. For the Raman and FT-IR measurement, the HSQ film was spin-coated on the Si wafers, and irradiated by the fabrication system with reflected light illumination system (Supplementary Fig. 1b).

**Measurements**. The morphology and shape of HSQ microstructures constructed by FsLDW were measured by using field-emission scanning electron microscope (FE-SEM, S-4800, Hitachi) at 5 kV accelerating voltage after being deposited with a thin layer of Au. The surface roughness of the HSQ microstructures were measured using an atomic force microscope (AFM, Fastscan IconBio, Bruker). The optical microscopy images were characterized by a laser scanning confocal microscope (LSCM, A1R MP, Nikon) using an oil-immersion objective lens (N.A. = 1.40, 60 ×, Nikon). Raman and FT-IR spectra of HSQ film with and without fs laser exposure were characterized by using Raman spectroscopy (Raman-11, Nanophoton, excitation wavelength is 532 nm) and FT-IR spectroscopy (Vertex 70, Bruker) equipped with microscope (Hyperion 1000, Bruker). The reflectance spectra of the structural colour before and after thermal treatment were measured by using FT-IR spectroscopy (Vertex 70, Bruker) equipped with microscope (Hyperion 1000, Bruker). Structural colour of the HSQ microstructure was recorded by employing a charge coupled device (CCD) camera (Beijing Groupca Company) equipped on an optical microscope under reflection mode. The absorption spectrum of the HSQ film on quartz was recorded on a UV-Visible Spectrophotometer (UV-2550, Shimadzu).

**Optical characterization**. The focusing property of the HSQ Fresnel lens is characterized using a home-made experimental setup (Supplementary Fig. 14). The experimental setup includes a 532 nm CW laser, linear variable neutral density filters, a 3D precisely controlled stage, an objective lens, a CCD camera and a personal computer. The attenuated laser beam is perpendicularly irradiated on the HSQ Fresnel lens, and the focusing property of the specimen were evaluated and recorded using the CCD camera connected to the personal computer. The light intensity distribution along the x-axis was evaluated by using the ImageJ software.

HSQ gratings were characterized by illuminating the specimens with a CW green laser (532 nm, Shaan' xi Richeng Technology Development Co., Ltd) and the transmitted light was recorded on a flat screen, which is placed 10 cm away and perpendicular to the laser beam. The diffraction patterns were recorded by a mobile phone camera (Mi 6, Xiaomi Company). For the analysis of diffraction angle, a graph paper is used as the plate screen to accurately measure the position of the diffraction spots.

**Thermal and chemical resistance characterization**. For thermal stability measurement of the HSQ microstructures fabricated by FsLDW, 3D HSQ microstructure was heated at different temperature and atmosphere. Firstly, 3D HSQ microstructure was heated on a hotplate (HP-1010, Hanbang Company) at the temperature of 400 °C in air for 0.5 h. Then, the 3D HSQ microstructure was calcined in a tube oven (OTF-1200X, Hefei Kejing Company) under Ar atmosphere for 2 h at the temperature of 500, 600, and 700 °C, respectively. After heating at each temperature, SEM images of the 3D microstructure were collected and compared to evaluate the thermal stability of the HSQ microstructure. The thermal resistance of the HSQ Fresnel lens was assessed by characterizing the focusing property before and after heating it at 400 °C in air for 0.5 h. Moreover, to evaluate the chemical resistance of the HSQ microstructures after thermal treatment, we characterized the focusing property of the calcined HSQ Fresnel lens after being exposed to several typical chemical reagents for 1 h. The chemical reagents include toluene, chloroform, n-hexane, acetone, tetrahydrofuran, 2-propanol, hydrochloric acid, sulphuric acid ($H_2SO_4$, 98%), ammonia solution (25%), sodium hydroxide (0.1 M aqueous solution), and hydrogen peroxide, respectively.

## Data availability

The data that support the findings of this study are available from the corresponding authors upon request.

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

## Acknowledgements

The authors thank the financial support of the National Key R&D Program of China (Grant Nos. 2016YFA0200501 and 2016YFC1100502), National Natural Science Foundation of China (NSFC, Grant Nos. 51673208, 51473176 and 61975213), Beijing Natural Science Foundation (Grant No. 2182079), and International Partnership Program of Chinese Academy of Sciences (GJHZ2021130). The authors thank Prof. Peng Xi for valuable discussion about the AFM images.

## Author contributions

F. Jin designed the experiments, fabricated and characterized the sample with assistance from Y.Y. Zhao and X.Z. Dong, and wrote the paper. F. Jin and J. Liu performed the Raman and FT-IR measurement. Y.Y. Zhao performed the theoretical analysis and Comsol simulations. M.L. Zheng supervised the project and wrote the paper. X.M. Duan supervised the project and wrote the paper. All authors discussed the results and contributed to writing the manuscript.

## Competing interests

The authors declare no competing interests.
