## [Peer Review File · Nature Communications]

Reviewers' comments:

Reviewer #1 (Remarks to the Author):

GENERAL: The manuscript is indeed very interesting, present novel original findings and of currently very hot topic, but the results are rather a collection of incremental measurements and the presentation is not adequate to clearly express it. Anyhow, the paper is a significant advance in the field which will positively influence the growth of the field itself.

MAJOR CONCERNS:

1. First of all, the title is very impressive claiming " $\lambda/30$ Inorganic Features Achieved..", but this $\lambda/30$ is shown in a very restricted way, making some features which are hardly reproducible and of no practical use (even the 53 nm line-width is so non-uniform, that it is compromising to call it a line; whereas the 26 nm feature is induced due to shrinkage and is obtainable just in the middle of, again, so called line). On the other hand, the fact that inorganic photo-resist HSQ is 3D structured via a multi-photo lithography method is really a great technological advance in material processing sense. The functional micro-optical prototypes do not employ the highest achieved resolution, rather just the fact of 3D high resolution free-form structuring. Thus, as a Reviewer I would recommend to rearrange the story so, that the main impact is the free-form direct laser structuring of inorganic resin and stress the enhanced functions (measured properties) of the additively manufactured objects.
2. There is no "nice/true" free-form/3D nanostructure shown among all the SEM images. The dots, lines are 2D or 2.5D structures at most. And the double layer 3D structure is just 2-translated layers, yet also of very poor structural quality. If it's a true 3D lithography, so a true 3D structures should be shown. All the demonstrated functions are of 2D line based structures, thus takes completely no advantage of the deep sub-wavelength features or the proposed 3D direct laser structuring potential of inorganic resin.
3. The fact that after heat-treatment the features swell instead of shrink is contra-intuitive. On the other hand, it still can be exploited for some specific applications like micro-fluidics or micro-mechanics (perhaps even more), yet the chosen application is micro-optics where it gives no advantage at all. Thus, basically the technological novelty of the shown approach is not serving for the proposed application at all and makes the paper look like a merged of several independent incremental studies instead of a single breakthrough.
4. For several times it is claimed how beneficial it is to have the non-photosensitized resin in the 3D printed nanostructures. Completely agreed on it, however some recent research is already published regarding this question, see: "Optically Clear and Resilient Free-Form μ -Optics 3D-Printed via Ultrafast Laser Lithography, Materials 2017, 10(1), 12; <https://doi.org/10.3390/ma10010012>". Thus it does not contradict the claim completely, but rather reduces its novelty and needs to be synchronized with the already reported relevant studies.
5. Why so slow writing speeds are employed of just 1-20 $\mu\text{m/s}$? Recent reports demonstrate much higher throughput (up to 10 mm/s in a routine fashion) for multiphoton 3D lithography, for instance: "Mesoscale laser 3D printing, Optics Express 27(11), 15205-15221 (2019) <https://doi.org/10.1364/OE.27.015205> . Should be considered and explained of the chosen parameters in respect to the state-of-the-art ones. Even the most standard equipment allows scanning linear speed of at least 100 $\mu\text{m/s}$. By the way, what are the speeds for the e-beam lithography (at least comparing to the one provided in the references [34-37])? It seems that the "simple and robust" method has no technological advantage for the introduced study and proposed potential applications.

6. Regarding the mechanisms determining the reaction type in the ultrafast laser 3D lithography there is a classical paper revealing it, namely studying the avalanche/ionization mechanisms as also discussed by the Authors, yet mentioning less important papers for the employed additive manufacturing case: Mechanisms of three-dimensional structuring of photo-polymers by tightly focussed femtosecond laser pulses, Optics Express 18(10), 10209-10221 (2010); <https://doi.org/10.1364/OE.18.010209>.

7. One more relevant and recent paper dealing with multi-photon 3D lithography and thermal-treatment is not mentioned, though very closely related to the described findings: Beyond 100 nm resolution in 3D laser lithography — Post processing solutions, Microelectronic Engineering 191, 25-31 (2018); <https://doi.org/10.1016/j.mee.2018.01.018> .

MINOR:

1. Page 6, line 121: “.even smaller feature size could be achieved by approaching closer to energy threshold.” – first of all, it’s the intensity, but not the energy is determining the threshold (as claimed by Authors as well). The question is – what prohibited to demonstrate the successful approach towards to the estimated threshold?
2. Why the linewidth on substrate is 53 nm and suspended just 26 nm? Is it an artefact of shrinkage and substrate restricting it? If so, the highest claimed resolution/feature sizes or of very limited application as are obtained indirectly, rather than in a controlled manner.
3. If the commercial glass started to melt at the elevated temperatures and limited the experimental conditions, why the mentioned quartz or sapphire substrates were not used as it was before in the already referenced literature where temperatures up to 1200 deg were applied: [21].

TECHNICAL QUESTIONS:

What is the refractive index (n) of the HSQ resin, especially as structured via multiphoton 3D lithography? And what is its deviation after/during heating? It is an essential parameter for manufacturing micro-optics, yet not measured or even evaluated in the manuscript.

TYPOS:

colud – could;
fromation – formation;
plat – flat.

Reviewer #2 (Remarks to the Author):

Major remarks:

- 1.) The authors mention that, the monomer did not exhibit any absorption for wavelength range between 800 nm and 193 nm and consequently the absorption occurs only by a multiphoton process at 157nm. Please prove that the process is a five-photon absorption as stated in the manuscript.

Conceptual experiments were shown in <https://doi.org/10.1002/adma.201402366> .

2.) AFM images shown in the suppl. Fig. 6 e are of low resolution. Please provide high-resolution AFM images of the (single-, cross-scanning) lines to prove the SEM results. Advantage: AFM does not require metallic coating and you could provide images of lines showing its FWHM. Furthermore, the low resolution AFM image of the lines does not really provide adequate height information. I believe a better resolution of the AFM-image is required to show this. Has an AFM-tip deconvolution been performed?

3.) Since the monomer requires a spin coating and a post baking, which limits the size of the written features and most likely introduces aberrations in the point spread function. What is the maximal thickness of the layer, which can be spin coated and structured? Can any aberrations for structures, which are further away from the surface be observed? Furthermore, authors claim that: 'Magnified SEM image indicates that the linewidth of the upper layer grid is smaller than that of the bottom layer, which is probably due to the shrinkage of the nanowires in the development process.' There could be multiple reasons - a post-polymerization shrinkage or aberration of the PSF, surface proximity effects induced by the non-Crosslinked HSQ oligomer or surface effects described by <https://doi.org/10.1111/j.1365-2818.2008.01940.x>. Please comment on that.

4.) Figure 2 is not representative – It is not evident why lines with 53nm and 26nm lateral feature size are depicted in one image. Typically, best results – in this case 26 nm - are demonstrated. Are the 26nm the FWHM of the line? The figure requires a better description.

5.) Figure 5 f – the quality of the AFM image could be improved. Could the authors comment on the low surface roughness measured ? Is it because of the homogenous spin-coating or is it insufficient washing? What was the spacing of the PSF's during the writing process? I would expect to see patterns induced by the PSF's.

6.) Fig. 6 represents 3D structures of polymers before thermal treatment. The lateral/axial resolution showed in this figures – minimal distance achieved between two lines is 149nm lateral and 439nm axial - are these the best achievable resolutions? Can you further improve the resolution?

Minor:

1.) 'Nonetheless, the feature size of inorganic microstructures obtained by 3D printing technology has been generally limited to a few micrometers, which is unsuitable for the applications requiring nanoscale resolution. ' - In this context, I believe this work is worth to be mentioned:

<https://doi.org/10.1021/acsnano.5b05863>

2.) Minor editing errors - e.g. line 86

3.) Figure 7 f – 'Full-width at half maximum (FWHM) of the optical intensity profile along x-axis of the focusing spot of HSQ Fresnel lens after being exposed to different chemical reagents – the authors mean the structure has been exposed to chemical reagents upon development or 'just due to imitate harsh' conditions? Not clear stated in the figure caption.

4.) The scale bars and the scales indicated arrows and in the zoomed out images frequently do not match in size.

Reviewer #3 (Remarks to the Author):

This paper demonstrates femtosecond laser direct writing (FsLDW) of hydrogen silsesquioxane (HSQ) through multi-photon absorption (MPA) processing for fabricating 2D and 3D nanoscale inorganic features. 26 nm feature size is achieved, which is about $\lambda/30$ of the wavelength of the exposure source and breaks the optical diffraction limit. The 3D HSQ microstructures shows excellent thermal stability up to 600C. The finding is exciting and has potential to fabricate inorganic micro-nano devices. However, the results have not reached the firm level to justify publication in Nature Communications, so I do not recommend to accept the paper for publication in this high-standard journal. The reasons are elaborated below:

1. The conceptual innovation of the paper is not significant. The fabrication of 3D HSQ features has been reported via electron-beam lithography [Nanotechnology 2016, 27, 254002]. The linewidth and spacing are almost equal to those in this paper. Moreover, photolithography has been proved to realize a HSQ structure with a 10 nm feature size [Journal of Vacuum Science & Technology B 2017, 35, 021603]. Finally, the authors claimed that inorganic features were fabricated by the MPA of the HSQ thin film, but the evidences were missed, including the MPA coefficients, the Z-scan plot and the PL spectra.
2. The fabrication mechanism of the ultrathin lines is not explained clearly. The authors proposed that the cross-scanning method could get the ultrafine feature size of 26 nm, because the additional line scribing (X1 and X2) could contribute to the HSQ polymerization. However, the contribution which is described as $2 \cdot a \cdot p \cdot 10^5 \cdot (e^{-2 \cdot (250/328)^2})^5$, is less than 0.03ap. The authors also claimed that the fluctuation of the fs laser intensity is $\pm 1\%$. The variation of the polymerization degree is more than 0.21ap (eq12, Supplementary Information), at least 7 times as much as the cross-scanning contribution. It is likely that the 26 nm size is generated from the laser fluctuation, rather than the crossing structure.
3. Regarding Table S1, how many samples did the authors count? A broad distribution of the hole/line sizes can be observed through Figure 6. In this case, the distribution is not negligible in the simulation of the feature size (X1 and X2 in eq 7, Supplementary Information).
4. Regarding Figure 3e, slower scanning speed means higher energy input to a fixed area on the HSQ film, but why does the slower speed lead to a higher damage threshold?
5. Supplementary Fig. 4f (Line 115, Page 6) was missed.

Point-by-point Response to the Reviewers' Comments

We thank the reviewers for their valuable comments and careful reading of our manuscript. Their thoughtful comments and suggestions have greatly helped us improve the quality of our manuscript. The following is our point-by-point response to the reviewers' comments.

Reviewers' comments:

Reviewer #1 (Remarks to the Author):

GENERAL: The manuscript is indeed very interesting, present novel original findings and of currently very hot topic, but the results are rather a collection of incremental measurements and the presentation is not adequate to clearly express it. Anyhow, the paper is a significant advance in the field which will positively influence the growth of the field itself.

Our reply: We thank the reviewer for careful reading and positively assessing our manuscript. We appreciate the valuable comments and suggestions, which will greatly improve the quality of the manuscript.

MAJOR CONCERNS:

1. First of all, the title is very impressive claiming “ $\lambda/30$ Inorganic Features Achieved..”, but this $\lambda/30$ is shown in a very restricted way, making some features which are hardly reproducible and of no practical use (even the 53 nm line-width is so non-uniform, that it is compromising to call it a line; whereas the 26 nm feature is induced due to shrinkage and is obtainable just in the middle of, again, so called line). On the other hand, the fact that inorganic photo-resist HSQ is 3D structured via a multi-photo lithography method is really a great technological advance in material processing sense. The functional micro-optical prototypes do not employ the highest achieved resolution, rather just the fact of 3D high resolution free-form structuring. Thus, as a Reviewer I would recommend to rearrange the story so, that the main impact is the free-form direct laser structuring of inorganic resin and stress the enhanced functions (measured properties) of the additively manufactured objects.

Our reply: We thank the reviewer for the thoughtful comments. We agree with the reviewer that $\lambda/30$ is shown in a restricted way. In our manuscript, this (26 nm, $\lambda/30$) is achieved by employing cross-scanning method through multi-photon lithography. Nonetheless, we think that these features are reproducible under optimized conditions and of practical use. The

Figure Response 1. **a-1** Tilted SEM image of 2D grid fabricated by FsLDW employing cross-scanning method. **a-2** Magnified SEM image of the region marked by red rectangle in a-1. The inset is the tilted SEM image of the HSQ features. **a-3** Magnified SEM image of the region marked by green rectangle in a-1. The inset is the tilted SEM image of the HSQ feature. **b-1** SEM image of 3D HSQ double layered grid fabricated by FsLDW employing cross-scanning method. **b-2** Magnified SEM image of the region marked by red rectangle in b-1. The inset is the tilted SEM image of the HSQ feature. **b-3** Magnified SEM image of the region marked by green rectangle in b-1. The inset is the tilted SEM image of the HSQ features

reasons are listed as follows. *Firstly*, the reproducible construction of 26 nm feature is feasible. By using cross-scanning method, we have fabricated nanowires with 26 and 28 nm in the 2D HSQ grid, and nanowires with 26 and 33 nm in the 3D HSQ grid, as shown in Figure Response 1 (Supplementary Fig. 5 in the revised supporting information). *Secondly*, the 53 and 26 nm HSQ nanowires are

freestanding features on the substrate, which means that the shrinkage is limited by the substrate. We believe that shrinkage is not the dominant formation mechanism for the 26 nm features, although it may occur during the development process. *Thirdly*, the quality of the 26 nm feature is good, with smooth edge. *Finally*, the nanowires with nanoscale feature size fabricated by the cross-scanning method could find useful applications, for example, 3D photonic crystals with photonic bandgap in NIR and even visible range. Furthermore, HSQ nanowire with the narrowest linewidth of 33 nm is achieved by employing the scanning speed of 100 $\mu\text{m/s}$, as shown in Fig. 2c in the revised manuscript. The quality of the 33 nm HSQ nanowire is as good as the features obtained by organic photoresist.

As mentioned by the reviewer, “the fact that inorganic photo-resist HSQ is 3D structured via a multi-photon lithography method is really a great technological advance in material processing sense.” To depict the enhanced functions of the 3D HSQ features by multi-photon lithography, we have fabricated 3D woodpile microstructure (Supplementary Fig. 7) and the biomimetic structural colour with thermal resistance (Fig. 6) through the FsLDW of 3D HSQ microstructures, and added the discussion in the revised manuscript (L270-L303).

2. There is no “nice/true” free-form/3D nanostructure shown among all the SEM images. The dots, lines are 2D or 2.5D structures at most. And the double layer 3D structure is just 2-translated layers, yet also of very poor structural quality. If it’s a true 3D lithography, so a true 3D structures should be shown. All the demonstrated functions are of 2D line based structures, thus takes completely no advantage of the deep sub-wavelength features or the proposed 3D direct laser structuring potential of inorganic resin.

Our reply: We thank for the reviewer for the constructive comments. We use “3D” in the title due to two reasons. *Firstly*, multi-photon lithography is intrinsically able to fabricate 3D microstructures. *Secondly*, we depicted 3D microstructure in Fig. 4g-i, i.e. double layered grid microstructures. We did not show more true free-form/3D structures in the previous manuscript, because the thickness of the HSQ film (XR-1541-004, Dow Corning) is very thin by spin-coating, about 80 nm at 4000 rpm for 60 s. The thin thickness hinders us from fabricating complex 3D nanostructures. During the fabrication of 3D microstructures shown in the previous manuscript, we obtained the thicker HSQ films by decreasing the spin-coating speed to 1000 rpm and repeating the spin-coating three times. However, the HSQ films are insoluble in the developer even without laser exposure if we repeated the spin-coating for four times and more, which hinders us from getting much thicker HSQ film and more complex 3D microstructures.

We agree with the reviewer that we should exhibit more free-form/3D nanostructures or take advantage of the deep sub-wavelength features. To achieve that, we purchased another HSQ photoresist (FOX-16, Dow Corning) to make HSQ films with thicker thickness, and have achieved 3D woodpile microstructure by using the FOX-16 photoresist, as shown in Fig. Response 2 (Supplementary Fig. 7 in the revised supporting information). Accordingly, we added the discussion in the revised manuscript “3D woodpile microstructure has also been fabricated by

using FOX-16, which can make thicker HSQ film (Supplementary Fig. 7).”

Figure Response 2. a Tilted SEM image and **b** Magnified tilted SEM image of 3D HSQ woodpile microstructure fabricated by FsLDW using FOX-16

3. The fact that after heat-treatment the features swell instead of shrink is counter-intuitive. On the other hand, it still can be exploited for some specific applications like micro-fluidics or micro-mechanics (perhaps even more), yet the chosen application is micro-optics where it gives no advantage at all. Thus, basically the technological novelty of the shown approach is not serving for the proposed application at all and makes the paper look like a merged of several independent incremental studies instead of a single breakthrough.

Our Reply: We thank the reviewer for the comments. We agree with the reviewer that HSQ features will shrink after heat-treatment, which we have discussed in the previous manuscript. The swell of the HSQ nanowire was probably attributed to the electron beam scanning during the SEM characterization under high magnification. Nevertheless, this does not depict the structural change of the HSQ microstructures after thermal treatment. In order to demonstrate the shrinkage of the HSQ double-layer grid after heat treatment, we evaluate the change of horizontal size of the HSQ double-grid under different temperature. As shown in Fig. 5 (b)-(b'') in the revised manuscript, the shrinkage of the HSQ microstructure is evaluated to be about 6.7% after thermal treatment at 600 °C. Accordingly, we revised Fig. 5 and changed the discussion in the revised manuscript from “Magnified SEM images in Fig. 5b, b', b'' and b''' show that the linewidth of the selected nanowire increases from 88 nm (pristine), to 92 nm (400 °C), 103 nm (500 °C), and 114 nm (600 °C), respectively.” to “SEM images in Fig. 5b, b', b'' and b''' show that the periodicity in horizontal direction decreases from 1008 nm (pristine), to 1005 nm (400 °C), 1001 nm (500 °C), and 940 nm (600 °C), respectively.”

As an inorganic photoresist, cured HSQ could be transferred to silica, i.e. glass. Glass has numerous applications, including micro-fluidics, micro-mechanics and optics. Especially, high quality glass is a significant driving force to facilitate the development of optics. Recently, researchers have achieved hybrid photonic integration by using 3D printing of polymeric photoresist¹. Moreover, 3D photonic crystals have been fabricated by using polymeric photoresist². Compared with the organic photoresists, inorganic photoresist HSQ possesses the merit of temperature and chemical tolerance. As a result, inorganic features will be an

emerging and urgent method for micro-optics. In the manuscript, we just exhibit the preliminary application of inorganic feature in micro-optics. In the future, we will further explore the applications in micro-fluidics and micro-mechanics.

References

1. Dietrich P.-I., Blaicher, M., Reuter, I., Billah, M., Hoose, T., Hofmann, A., Caer, C., Dangel, R., Offrein, B., Troppenz, U., Moehrle, M., Freude, W., & Koos, C., In situ 3D nanoprinting of free-form coupling elements for hybrid photonic integration. *Nat. Photonics* 12, 241-247 (2018).

2. Liu, Y., Wang, H., Ho, J., Ng, R. C., Ng, R. J., Hall-Chen, V. H., Koay, E. H., Dong, Z., Liu, H., Qiu, C., Greer, J., & Yang, J. K., Structural color three-dimensional printing by shrinking photonic crystals. *Nat. Commun.* 10, 4340 (2019).

4. For several times it is claimed how beneficial it is to have the non-photosensitized resin in the 3D printed nanostructures. Completely agreed on it, however some recent research is already published regarding this question, see: “Optically Clear and Resilient Free-Form μ -Optics 3D-Printed via Ultrafast Laser Lithography, *Materials* 2017, 10(1), 12; <https://doi.org/10.3390/ma10010012>; Thus it does not contradict the claim completely, but rather reduces its novelty and needs to be synchronized with the already reported relevant studies.

Our Reply: We thank the reviewer for the constructive comments. As mentioned by the reviewer, there are some publications on non-photosensitized resin in the 3D printed nanostructures. This suggests that more and more researchers have noticed the beneficial merits of the non-photosensitized resins. Nevertheless, we think that the reported studies would not reduce the novelty of the manuscript. *Firstly*, HSQ is an inorganic resin, while the reported non-photosensitized resins in the 3D fabrication are mainly organic or organic/inorganic hybrid resins. The organic component in the organic and hybrid resins endow the resins with low thermal and mechanical resistance. Extra thermal treatment is needed for the hybrid resins, and obvious shrinkage will occur during the thermal treatment. By contrast, inorganic features fabricated by HSQ in the manuscript exhibit thermal and chemical resistance. *Secondly*, the feature size of the freestanding HSQ microstructure is 26 nm, which is much smaller than the reported results. The result is significantly important and novel because the finer microstructures are pursued in the 3D fabrication. Moreover, HSQ is prospective for constructing both micro-electronic and micro-optics devices, while the reviewer mentioned paper only demonstrates the potential applications in micro-optics devices. Thus, we think the relevant published studies do not reduce the novelty of the manuscript. Nevertheless, we add the reported study as the reference in the revised manuscript to emphasize the importance of non-photosensitized resin in the 3D printed microstructures. Accordingly, we revised the manuscript as follows: ‘Analogous to the organic photoresist, hybrid organic/inorganic photoresist SZ2080 has been fabricated by employing FsLDW without the addition of photoinitiator^{24, 25}.’

25. Jonušauskas, L., Gailevičius, D., Mikoliunaitė, L., Šakalauskas, D.,

Sakirzanovas, S., Juodkazis, S., & Malinauskas, M., **Optically clear and resilient free-form μ -Optics 3D-Printed via ultrafast laser lithography.** *Materials* 10(1), 12(2017).

5. Why so slow writing speeds are employed of just 1-20 $\mu\text{m/s}$? Recent reports demonstrate much higher throughput (up to 10 mm/s in a routine fashion) for multiphoton 3D lithography, for instance: “Mesoscale laser 3D printing, *Optics Express* 27(11), 15205-15221 (2019) <https://doi.org/10.1364/OE.27.015205> . Should be considered and explained of the chosen parameters in respect to the state-of-the-art ones. Even the most standard equipment allows scanning linear speed of at least 100 $\mu\text{m/s}$. By the way, what are the speeds for the e-beam lithography (at least comparing to the one provided in the references [34-37])? It seems that the “simple and robust” method has no technological advantage for the introduced study and proposed potential applications.

Our Reply: We thank the reviewer for the thoughtful comments. We choose the slow writing speed due to two reasons. *Firstly*, HSQ features with very small feature size can be obtained by FsLDW at slow writing speeds. For femtosecond laser lithography, microstructures with superior feature sizes could be obtained by the combination of high speed and high laser intensity, or slow speed and low laser intensity. By employing the writing speed of 5 $\mu\text{m/s}$, HSQ nanowires with 26 nm feature size can be readily achieved via cross-scanning method. *Secondly*, the fabrication window of HSQ by FsLDW at slow writing speed is broader than that at high writing speeds. Due to the low sensitivity of HSQ by FsLDW, the photopolymerization threshold and damage threshold is close, resulting in the narrow fabrication window. Consequently, slow writing speeds are employed to perform FsLDW of HSQ in the previous manuscript.

Figure Response 3. SEM image of HSQ nanowires fabricated by FsLDW with the scanning speed from 5 to 100 $\mu\text{m/s}$, and magnified SEM image of HSQ nanowire by FsLDW with scanning speed of 100 $\mu\text{m/s}$

We agree with the reviewer that high writing speed is critically important for multi-photon 3D lithography by improving the efficiency for higher throughput. Fortunately, we could further improve the writing speed of HSQ by FsLDW. By optimizing the fs laser intensity, we have obtained HSQ nanowires by increasing

the laser scanning speed from 5 to 100 $\mu\text{m/s}$, as shown in Fig. Response 3 (Fig. 2c in the revised manuscript). The HSQ nanowires fabricated with different writing speeds exhibit high resolution and good quality. 33 nm HSQ nanowire can be achieved by FsLDW with the scanning speed of 100 $\mu\text{m/s}$.

Multi-photon lithography is simple and robust in fabricating 3D HSQ features compared to the methods depicted in the references [34-37] (original reference 34-37, i.e. reference 38-41 in the revised manuscript), in which the methods need complex process and long time to fabricate 3D HSQ microstructures. For example, 9 steps and precise alignment are needed to fabricate 3D microstructures with two layers (original reference 34, i.e. reference 38 in the revised manuscript). Although the authors of the references did not mention the scanning speed for the 3D microstructures, 4 hours/layer are required to construct 3D microstructures (original reference 37, i.e. reference 41 in the revised manuscript). In our method, only 1 hour is required to fabricate a double-layer 3D HSQ microstructure even with the scanning speed of 5 $\mu\text{m/s}$. Much shorter time will be needed to fabricate 3D HSQ microstructures if we further increase the scanning speed to 100 $\mu\text{m/s}$. As a result, we believe the multi-photon lithography of HSQ is a simple and robust method to fabricate 3D inorganic features.

6. Regarding the mechanisms determining the reaction type in the ultrafast laser 3D lithography there is a classical paper revealing it, namely studying the avalanche/ionization mechanisms as also discussed by the Authors, yet mentioning less important papers for the employed additive manufacturing case: Mechanisms of three-dimensional structuring of photo-polymers by tightly focussed femtosecond laser pulses, *Optics Express* 18(10), 10209-10221 (2010); <https://doi.org/10.1364/OE.18.010209>.

Our Reply: We thank the reviewer for the constructive comment. We are sorry we omitted the paper as the reviewer mentioned, during discussing the mechanism of the multi-photon lithography of HSQ features. We have added the reference and discussion in the revised manuscript as follows. “Especially, with very high laser intensity, fs laser pulse is capable of directly breaking the chemical bond of monomer molecules and triggering the crosslinking of organic photoresists without any photoinitiators via multi-photon ionization (MPI) process^{22,23}.”
23 Malinauskas, M., Žukauskas, A., Bičkauskaitė, G., Gadonas, R. & Juodkazis, S., Mechanism of three-dimensional structuring of photo-polymers by tightly focussed femtosecond laser pulses. *Opt. Express* 18(10), 10209-10221(2010).

7. One more relevant and recent paper dealing with multi-photon 3D lithography and thermal-treatment is not mentioned, though very closely related to the described findings: Beyond 100 nm resolution in 3D laser lithography — Post processing solutions, *Microelectronic Engineering* 191, 25-31 (2018); <https://doi.org/10.1016/j.mee.2018.01.018>.

Our Reply: We thank the reviewer for the constructive comment. As discussed in

the previous manuscript, we depict the multi-photon 3D lithography of sub-100 nm inorganic features, which exhibit thermal and chemical resistance. In the paper the reviewer mentioned, the authors demonstrated multi-photon 3D lithography of organic photoresist, which can be transferred to microstructures with sub-100 nm resolution after thermal and/or plasma etching treatment. There are some differences between the result in the paper and our work. *Firstly*, we can directly fabricate sub-100 nm features by FsLDW. However, the authors did not directly construct sub-100 nm features, instead obtained sub-100 features by post thermal and/or plasma treatment. *Secondly*, the inorganic features in our work possess thermal resistance, while the features demonstrated in the paper exhibit low thermal stability due to the thermal decomposition. Although there are some obvious differences between the paper and our manuscript, we still believe it would give a clue to achieve finer nanostructures by employing the post treatment method discussed in the paper. Consequently, we add the paper into the references in the revised manuscript as follows. “Through pyrolysis of 3D features constructed by 3D printing of photoresists containing inorganic precursors and/or nanoparticles, various 3D inorganic features have been successfully achieved⁷⁻¹¹.”.

11. Seniutinas, G., Weber, A., Padeste, C., Sakellari, I., Farsari, M., & David, C. Beyond 100 nm resolution in 3D laser lithography-Post processing solutions. *Microelectronic Engineering* 191, 25-31(2018).

MINOR:

1. Page 6, line 121: “..even smaller feature size could be achieved by approaching closer to energy threshold.” – first of all, it’s the intensity, but not the energy is determining the threshold (as claimed by Authors as well). The question is – what prohibited to demonstrate the successful approach towards to the estimated threshold?

Our Reply: We thank the reviewer for the comments. We have corrected the mistake as follows. “..even smaller feature size could be achieved by approaching closer to laser intensity threshold. ” We think that the difficulty in prohibiting the successful approach towards to the threshold lies in the following experimental parameters: the precise controlling of the laser intensity (the accuracy of the laser intensity and the fluctuation of the laser intensity), the position of the laser focus relative to the substrate, and the scanning speed.

2. Why the linewidth on substrate is 53 nm and suspended just 26 nm? Is it an artefact of shrinkage and substrate restricting it? If so, the highest claimed resolution/feature sizes or of very limited application as are obtained indirectly, rather than in a controlled manner.

Our Reply: We thank the reviewer for the comments. The linewidth of 53 nm and 26 nm are both achieved on the substrate, and none is suspended feature. The linewidth of 53 nm is obtained by employing single-scanning method, while that of 26 nm is achieved via cross-scanning method. We compared the linewidth of the nanowires on the substrate to study the best feature size by using different scanning methods. Generally, the feature size on substrate depicts the real and direct capability of multi-photon lithography, since the substrate restricts the shrinkage during the lithography and/or developing process. By contrast,

suspended microstructures are obtained indirectly, in which the shrinkage would significantly reduce the feature size in an uncontrollable manner. We agree with the reviewer that suspended features have very limited application since they are obtained indirectly in an uncontrolled manner. However, the feature size of 26 nm on substrate is critically important, which can be obtained directly in a controlled manner in our manuscript. To avoid the misunderstanding, we add Supplementary Fig. 5 and the discussion in the revised manuscript to emphasize the fact that the freestanding 26 nm HSQ feature size is obtained on the substrate. “Freestanding HSQ nanowire on the substrate with the narrowest linewidth of 26 nm has been successfully constructed (Fig. 2b, Supplementary Fig. 4b, and Fig. 5), which is only $\lambda/30$ of the 780 nm fs laser.”

3. If the commercial glass started to melt at the elevated temperatures and limited the experimental conditions, why the mentioned quartz or sapphire substrates were not used as it was before in the already referenced literature where temperatures up to 1200 deg were applied: [21].

Our Reply: We thank the reviewer for the comments. We select commercial glass as the substrate since it is suitable for the multi-photon lithography of HSQ. *Firstly*, the commercial glass is transparent in the visible and near infrared (NIR) range, which makes it a suitable substrate for the construction of HSQ microstructures by FsLDW using NIR fs laser. *Secondly*, the commercial glass is employed due to the refractive index matching. The refractive index of the oil (for the oil-immersion objective lens), glass, and HSQ is about 1.516, 1.52 and 1.41, respectively. When the laser beam is focused through oil, glass into HSQ, the distortion of the focus spot is low due to the refractive index matching. However, the refractive index mismatching for quartz or sapphire substrate will be a problem during the focusing of the laser beam, especially for sapphire substrate (refractive index is about 1.777). *Thirdly*, the commercial glass substrate is of low-cost and convenient to get. As a commonly used substrate, glass substrate is cheap and convenient to obtain. As a result, we select the commercial glass as the substrate during the multi-photon lithography of HSQ. However, the reviewer’s suggestions give us a glue to solve the problem of melting of the substrate under high temperature. We will try other substrates and show the detailed results in our future work.

TECHNICAL QUESTIONS:

What is the refractive index (n) of the HSQ resin, especially as structured via multiphoton 3D lithography? And what is its deviation after/during heating? It is an essential parameter for manufacturing micro-optics, yet not measured or even evaluated in the manuscript.

Our Reply: We thank the reviewer for the constructive comments. Thanks for the comment. We agree with the reviewer that the refractive index of HSQ is an essential parameter for manufacturing micro-optics. It’s important to detect the refractive index of the HSQ features via multi-photon 3D lithography and further thermal treatment. We measured the refractive index (n) of the HSQ resin, HSQ features structured via multi-photon 3D lithography, and its deviation after heating (400, and 600 °C) by using a thin film analyzer (F40-UV, Filmetrics, USA).

Figure Response 4. The refractive index of the HSQ film (black square), HSQ feature (red ball) structured via multi-photon lithography, and structured HSQ features (red ball) heated in Ar atmosphere for 2 hours at 400 and 600 °C, respectively.

Figure Response 4 (Supplementary Fig. 9) plots the refractive index of HSQ resin, structured HSQ via fs laser direct writing, and structured HSQ after thermal treatment at 400 and 600 °C for 2 hours, respectively.

Accordingly, we added the description and discussion of the refractive index of HSQ resin, structured HSQ features, and structured HSQ features after thermal treatment in the revised supporting information as follows. “It’s important to detect the refractive index of HSQ features by fs laser lithography and further thermal treatment. We have measured the refractive index of the HSQ resin, HSQ features structured via fs laser direct writing, and its deviation after heating (400, and 600 °C) by using a thin film analyzer (F40-UV, Filmetrics, USA). The measurement procedure is briefly described as follows. *Firstly*, four HSQ films on Si wafer substrates were prepared by spin-coating and evaporation of the solvent. *Secondly*, HSQ micro-squares with the footprint of 50 μm x 50 μm were fabricated on Si wafer via FsLDW. To compare the effect of thermal treatment on the refractive index of the HSQ features, we prepared HSQ micro-squares on three Si substrates by using the same parameters. *Thirdly*, one HSQ micro-square was stored without further treatment after lithography, while another two HSQ micro-squares were treated in a tube oven in Ar atmosphere for 2 hours at the temperature of 400 and 600 °C, respectively. *Finally*, the refractive index of above samples at 632.8 nm were characterized by using F40-UV. Supplementary Fig. 9 plots the refractive index of HSQ resin, structured HSQ via FsLDW, and structured HSQ after thermal treatment in Ar atmosphere at 400 and 600 °C for 2 hours, respectively. The refractive index of HSQ film is 1.406, which is similar to

the reported value of 1.41². The refractive index of HSQ feature fabricated by FsLDW declines to 1.380. The decrease of refractive index is mainly attributed to the compositional change, i.e. the transfer of cage-network structure². After thermal treatment at 400 °C, the refractive index of structured HSQ features decreases from 1.380 to 1.355. The drop of the refractive index is ascribed to the film porosity under thermal treatment³. Further increase the temperature to 600 °C, the refractive index of the structured HSQ features increases from 1.355 to 1.375. The increase of the refractive index under higher temperature is mainly due to the disassociation of Si-H bond and film densification⁴. The evolution of the refractive index of HSQ by FsLDW and subsequent thermal treatment is significant for the construction of micro-optical devices.”

Supplementary References

2. Siampour, H., Kumar, S., Davydov, V. A., Kulikova, L. F., Agafonov, V. N., & Bozhevolnyi, S. I. On-chip excitation of single germanium vacancies in nanodiamonds embedded in plasmonic waveguides. *Light: Sci. Appl.* 7, 61(2018).
3. Yang, C. C. & Chen, W. C. The structures and properties of hydrogen silsesquioxane (HSQ) films produced by thermal curing. *J. Mater. Chem.* 12, 1138-1141 (2002).
4. Liou, H. C. & Pretzer J. Effect of curing temperature on the mechanical properties of hydrogen silsesquioxane thin films. *Thin Solid Film* 335, 186-191(1998).

TYPOS:

colud – could;

fromation – formation;

plat – flat.

Our Reply: We thank the reviewer for the constructive comments. We have checked the manuscript and corrected the mistakes.

Reviewer #2 (Remarks to the Author):

Major remarks:

1.) The authors mention that, the monomer did not exhibit any absorption for wavelength range between 800 nm and 193 nm and consequently the absorption occurs only by a multiphoton process at 157nm. Please prove that the process is a five-photon absorption as stated in the manuscript. Conceptual experiments were shown in <https://doi.org/10.1002/adma.201402366>.

Our Reply: We thank the reviewer for the constructive comments. We appreciate the reviewer’s suggestion on the in-site measurement of nonlinear absorption order during the FsLDW of HSQ by employing the method discussed in the

mentioned paper. We believe it will improve the quality of the manuscript if we directly confirm the order of the multi-photon absorption process by the measurement. So far, we do not have the experimental setup as shown in the reference, in which a fast, non-mechanical shutter is required. Besides, the fabrication window of HSQ by FsLDW is too narrow to support the conceptual experiment, in which a broad fabrication window is needed to confirm the order of the multi-photon process.

Accordingly, we changed the discussion in the revised manuscript from “Due to photocuring sensitivity of HSQ at 157 nm (7.89 eV)³⁰, effective five-photon absorption of HSQ is induced by 780 nm (1.59 eV) fs laser under high energy irradiation. Although no photoinitiator exists in HSQ, photocuring of HSQ is achieved by effective five-photon absorption triggered cleavage of Si-H bonds and subsequent crosslinking of HSQ, resulting in the fabrication of 3D nanoscale HSQ features.” to “Although no photoinitiator exists in HSQ, photocuring of HSQ is achieved by multi-photon absorption triggered cleavage of Si-H bonds and subsequent crosslinking of HSQ, resulting in the fabrication of 3D nanoscale HSQ features.”.

2.) AFM images shown in the suppl. Fig. 6 e are of low resolution. Please provide high-resolution AFM images of the (single-, cross-scanning) lines to prove the SEM results. Advantage: AFM does not require metallic coating and you could provide images of lines showing its FWHM. Furthermore, the low resolution AFM image of the lines does not really provide adequate height information. I believe a better resolution of the AFM-image is required to show this. Has an AFM-tip deconvolution been performed?

Our Reply: We thank the reviewer for the constructive comments. According to the reviewer’s suggestion, we measured AFM image of the HSQ lines with single-scanning method again, and obtained high-resolution AFM image of the HSQ lines. To demonstrate the height and linewidth information of the HSQ lines with nanoscale linewidth, high-resolution AFM images of HSQ nanowires with different scanning speeds were obtained, as shown in Supplementary Fig. 3 in the revised manuscript. Accordingly, we added Supplementary Fig. 3 and the discussion of the AFM images of the HSQ lines in the revised manuscript as follows. “AFM image depicts the height of 30 nm for the HSQ nanowire fabricated by the scanning speed of 100 $\mu\text{m/s}$ (Supplementary Fig. 3).”. Meanwhile, we delete the previous Supplementary Fig. 6e and the relevant discussion in the revised manuscript to avoid inaccurate results. Besides, we did not exhibit the AFM images of the cross-scanning lines due to the sample was destroyed by the thermal treatment at 700 $^{\circ}\text{C}$ (Supplementary Fig. 8m). We have not performed an AFM-tip deconvolution during the measurement.

3.) Since the monomer requires a spin coating and a post baking, which limits the size of the written features and most likely introduces aberrations in the point spread function. What is the maximal thickness of the layer, which can be spin coated and structured? Can any aberrations for structures, which are further away from the surface

be observed? Furthermore, authors claim that: ‘Magnified SEM image indicates that the linewidth of the upper layer grid is smaller than that of the bottom layer, which is probably due to the shrinkage of the nanowires in the development process.’ There could be multiple reasons - a post-polymerization shrinkage or aberration of the PSF, surface proximity effects induced by the non-Crosslinked HSQ oligomer or surface effects described by <https://doi.org/10.1111/j.1365-2818.2008.01940.x>. Please comment on that.

Our Reply : We thank the reviewer for the comments. The thickness of the photoresist layer is dependent on the spin coating speed for the HSQ (XR-1541-004, Dow Corning) used in this study. The maximal thickness of the HSQ layer is about 439 nm by spin coating three times, each time with 1000 rpm for 60 s.

The size of the written feature should be sensitive to the position of focus spot because the small thickness of the HSQ layer, and the refractive index mismatching between the HSQ layer and the air. Aberrations for structures can be observed when they are further away from the surface, as shown in Fig. 4. We analyzed the double-layer fishnet structures on the bottom edge of Fig. 4g, and found that the upper layer lines are all smaller than the bottom lines. We notice that they are all suspended lines, which means that none of them are restricted by the substrate. The difference is probably attributed to the different soaking time for the upper and bottom suspended HSQ nanowires during development process, because the developer will first contact the upper region of the HSQ film. Furthermore, the HSQ lines in the bottom layer are also different, in which the freestanding HSQ lines are bigger than the suspended ones. We think the difference between the freestanding HSQ lines and suspended lines in the bottom layer is mainly attributed to the different post-polymerization shrinkage between them. For the freestanding HSQ lines, the shrinkage is restricted by the substrate. By contrast, suspended HSQ lines are free from the restriction of the substrate, resulting in obvious shrinkage in the development process. We believe that the difference does not mainly come from the aberration of the point spread function due to the ultrathin thickness of the HSQ films.

Surface proximity effects induced by the non-crosslinked HSQ oligomer would also be a possible reason.

The surface effects described by <https://doi.org/10.1111/j.1365-2818.2008.01940.x>, mainly focus on the cured resin film as thin as several nanometer and even thinner on the substrate when the laser intensity is near the threshold. The polymeric lines in the study are uncontinuous and incomplete. Moreover, the study did not deal with multilayer microstructures.

Accordingly, we revised the manuscript from “Magnified SEM image indicates that the linewidth of the upper layer grid is smaller than that of the bottom layer, which is probably due to the shrinkage of the nanowires in the development process.” To “Magnified SEM image indicates that the linewidth of the upper layer grid is smaller than that of the bottom layer, which is probably due to the shrinkage of the nanowires in the development process and surface proximity effects induced by the non-crosslinked HSQ oligomer”.

4.) Figure 2 is not representative – It is not evident why lines with 53nm and 26nm lateral feature size are depicted in one image. Typically, best results – in this case 26 nm - are demonstrated. Are the 26nm the FWHM of the line? The figure requires a better description.

Our Reply: We thank the reviewer for the constructive comments. We depicted the 53 and 26 nm lateral feature size in one image in the previous manuscript because they are achieved by different scanning methods. The 53 nm is obtained by using single-scanning method, while the 26 nm is achieved by employing cross-scanning method. The 26 nm is the lateral size of the freestanding HSQ nanowire on the substrate, not the FWHM of the line. In the manuscript, we try to pursue the best feature size of HSQ microstructures by femtosecond laser direct writing, so we demonstrated the feature sizes with different methods. We further discussed the formation mechanism of 33 and 26 nm by different methods in the revised manuscript (from Line 143 to Line 160, Fig. 2a, and Supplementary Fig. 4). To avoid misunderstanding, we add more description in the revised manuscript as follows. ‘It’s worth noting that 33 nm and 26 nm HSQ features are both freestanding microstructures on the substrates, which are directly constructed by FsLDW with different scanning methods.’.

5.) Figure 5 f – the quality of the AFM image could be improved. Could the authors comment on the low surface roughness measured? Is it because of the homogenous spin-coating or is it insufficient washing? What was the spacing of the PSF’s during the writing process? I would expect to see patterns induced by the PSF’s.

Our Reply: We thank the reviewer for the comments. Due to metallic coating of the surface for SEM image measurement, it’s unsuitable to achieve high-quality AFM image for the micro-square. Thus, we have fabricated the HSQ micro-square again, and obtained the high-quality AFM image of the surface, as shown in Fig. 4f. The micro-square exhibits very smooth surface with a roughness of about 3 nm. The results verify that it’s reproducible to fabricate HSQ microstructure with low surface roughness by FsLDW. Accordingly, we replaced the Fig. 4f and changed the discussion in the revised manuscript as follows. “AFM measurement of the microstructure shows very smooth surface with a roughness of about 3 nm (Fig. 4f). The surface roughness of HSQ by FsLDW is comparable with that of the HSQ microstructures fabricated by E-beam lithography, and that of the sintered glass microstructures by stereolithography, as well^{7, 53}.”

The low surface roughness would be attributed to several reasons. *Firstly*, the photopolymerization property of the HSQ by FsLDW. The sensitivity of HSQ film to femtosecond laser is low. The small fluctuation of the laser intensity will not lead to obvious unflatness of the polymerized region. *Secondly*, the micro-square is fabricated by optimizing the laser parameters, such as laser intensity, scanning speed, and overlapping of the adjacent scanning. The surface roughness of the microstructures could be effectively reduced by employing small spacing between adjacent scanning of 100 nm. *Thirdly*, the thickness of the HSQ film is very small and the surface is flat after spin-coating. The thickness of HSQ films is about 80

nm, which is small compared to the size of the laser spot in vertical direction. The whole HSQ film could be polymerized under laser irradiation. Meanwhile, the uncured HSQ film could be very flat. The combination of small thickness and flat surface before curing will also lead to low surface roughness. However, the samples have been washed completely after laser irradiation. No residual unpolymerized resin could be observed on the substrate. We believe insufficient washing is not the reason to make the low surface roughness.

The spacing of the PSF's (of the adjacent lines) is 100 nm in the fabrication of HSQ micro-square by FsLDW. Accordingly, we performed the simulation of optical intensity distribution during the fabrication of HSQ micro-square by FsLDW, and the result is shown in Fig. Response 5. Three-dimensional optical intensity distribution depicts the uniform laser intensity along the surface (Fig. Response 5b). Furthermore, the top view and side view of the light intensity distribution also clearly demonstrate the uniform laser intensity during the FsLDW. Consequently, the uniform laser intensity distribution results in the low roughness of the HSQ micro-square by FsLDW.

Figure Response 5. **a** Scheme of the fabrication of micro-square by FsLDW with the spacing between the adjacent lines of 100 nm. **b** Simulated three-dimensional optical intensity distribution during the fabrication of HSQ micro-square by FsLDW. **c** Top view and **d** side view of the optical intensity distribution shown in **b**

6.) Fig. 6 represents 3D structures of polymers before thermal treatment. The lateral/axial resolution showed in this figures – minimal distance achieved between two lines is 149nm lateral and 439nm axial - are these the best achievable resolutions? Can you further improve the resolution?

Our Reply: We thank the reviewer for the comments. In Fig. 6 (Fig. 5 in the revised manuscript), the distance between two lines is set to 500 nm lateral and 439 nm axial. The 149 nm is the diameter of a hole in the 3D structures formed by FsLDW due to the expanding of the polymerized region through the cross-scanning

method. In this study, the smallest distance between two lines is 400 nm lateral, as shown in Fig. 4a. The question gives us a clue to further improving the resolution in our future work.

Minor:

1.) ‘Nonetheless, the feature size of inorganic microstructures obtained by 3D printing technology has been generally limited to a few micrometers, which is unsuitable for the applications requiring nanoscale resolution. ‘ - In this context, I believe this work is worth to be mentioned: <https://doi.org/10.1021/acsnano.5b05863>

Our Reply: We thank the reviewer for the comments. According to the suggestion of the reviewer, we added the reference in the revised manuscript. “Under fs laser irradiation, photocleavage of photoactive molecules and subsequent polymerization or cleavage will induce different solubility of the photoresists, resulting in the fabrication of well-defined microstructures¹⁹⁻²¹.”

19. Buchegger, B., Kreutzer, J., Plochberger, B., Wollhofen, R., Sivun, D., Jacak, J., Schütz, G. J., Schubert, U. & Klar, T. A. Stimulated emission depletion lithography with mercapto-functional polymers. *ACS Nano* 10, 1954-1959 (2016).

2.) Minor editing errors - e.g. line 86

Our Reply: We thank the reviewer for the constructive comment. We have corrected the editing errors.

3.) Figure 7 f – ‘Full-width at half maximum (FWHM) of the optical intensity profile along x-axis of the focusing spot of HSQ Fresnel lens after being exposed to different chemical reagents – the authors mean the structure has been exposed to chemical reagents upon development or ‘just due to imitate harsh’ conditions? Not clear stated in the figure caption.

Our Reply: We thank the reviewer for the comments. We demonstrate the chemical resistance of the HSQ microstructures by using different chemicals to imitate harsh conditions. To clear up the confusion, we have revised the figure caption to “Full-width at half maximum (FWHM) of the optical intensity profile along x-axis of the focusing spot of HSQ Fresnel lens after being exposed to different chemical reagents, which are selected to imitate the harsh conditions.”

4.) The scale bars and the scales indicated arrows and in the zoomed out images frequently do not match in size.

Our Reply: We thank the reviewer for the constructive comment. We have carefully checked the figures and corrected the mistakes in the revised manuscript.

Reviewer #3 (Remarks to the Author):

This paper demonstrates femtosecond laser direct writing (FsLDW) of hydrogen silsesquioxane (HSQ) through multi-photon absorption (MPA) processing for fabricating 2D and 3D nanoscale inorganic features. 26 nm feature size is achieved, which is about $\lambda/30$ of the wavelength of the exposure source and breaks the optical diffraction limit. The 3D HSQ microstructures shows excellent thermal stability up to 600 °C. The finding is exciting and has potential to fabricate inorganic micro-nano devices. However, the results have not reached the firm level to justify publication in Nature Communications, so I do not recommend to accept the paper for publication in this high-standard journal. The reasons are elaborated below:

Our Reply: We appreciate the positive assessment of our work. We will try our best to improve the quality of our manuscript to satisfy the standard of this high-rank journal.

1. However, the results have not reached the firm level to justify publication in Nature Communications, so I do not recommend to accept the paper for publication in this high-standard journal. The reasons are elaborated below: The conceptual innovation of the paper is not significant. The fabrication of 3D HSQ features has been reported via electron-beam lithography [Nanotechnology 2016, 27, 254002]. The linewidth and spacing are almost equal to those in this paper. Moreover, photolithography has been proved to realize a HSQ structure with a 10 nm feature size [Journal of Vacuum Science & Technology B 2017, 35, 021603]. Finally, the authors claimed that inorganic features were fabricated by the MPA of the HSQ thin film, but the evidences were missed, including the MPA coefficients, the Z-scan plot and the PL spectra.

Our Reply: We thank the reviewer for the comments. After careful evaluation, we think that the innovation of the revised paper is clear and significant. *Firstly*, although 3D HSQ features have been fabricated via electron-beam lithography [Nanotechnology 2016, 27, 254002], they could only construct simple microstructures by layer-by-layer stacking, which need complex and time-waste process. Moreover, alignment is a serious problem for the fabrication of fine 3D HSQ features by electron-beam lithography. *Secondly*, photolithography needs expensive and complex EUV equipment to fabricate HSQ features with 10 nm feature size [Journal of Vacuum Science & Technology B 2017, 35, 021603]. Compared to electron-beam lithography and EUV lithography, FsLDW is a convenient and simple way to achieve nanoscale features, especially for 3D HSQ features. Moreover, nanoscale microstructures have been achieved to obtain $\lambda/30$ features, indicating the breaking of optical diffraction limit by employing FsLDW. As a result, we think the innovation of the paper is clear and significant.

We depict that inorganic features are fabricated by the multi-photon absorption (MPA) of the HSQ, since HSQ does not absorb light in the wavelength range between 200 and 800 nm (Supplementary Fig. 2), which verifies that the HSQ can't be photocured through single photon process by using 780 nm fs laser exposure.

Thus, we suggest that inorganic features were fabricated by the MPA, instead of single-photon process of the HSQ thin film. We tried several measurements to find out the nonlinear order of the MPA process, but haven't obtained effective results. The detailed process is discussed as follows. *Firstly*, HSQ does not exhibit detectable photoluminescence (PL) under femtosecond laser exposure, indicating that PL method is not suitable for HSQ. *Secondly*, we have measured the MPA characteristic of HSQ solution using Z-scan method, but failed. The laser intensity threshold for the MPA of the solvent is lower than that of the HSQ molecules. We did not detect effective MPA of HSQ until the appearance of white light of the solvent induced by the femtosecond laser. *Thirdly*, we have performed the in-site measurement of the nonlinear absorption order of HSQ using the method proposed by Prof. Mueller (<https://doi.org/10.1002/adma.201402366>, *Advanced Materials* 26(2014), 6566-6571.), and did not obtain the effective results. In this method, the nonlinear absorption order is evaluated by measuring the polymerization threshold under different exposure time. We tried to measure the polymerization threshold at different exposure time by changing the scanning speed, but failed to get the required data due to the narrow fabrication window of HSQ under femtosecond laser exposure. Nevertheless, we will continue to explore the detailed information of the MPA process of HSQ by FsLDW in our future work.

2. The fabrication mechanism of the ultrathin lines is not explained clearly. The authors proposed that the cross-scanning method could get the ultrafine feature size of 26 nm, because the additional line scribing (X1 and X2) could contribute to the HSQ polymerization. However, the contribution which is described as $2 \cdot a \cdot \rho \cdot 10^5 \cdot (e^{-2 \cdot (250/328)^2})^5$, is less than $0.03ap$. The authors also claimed that the fluctuation of the fs laser intensity is $\pm 1\%$. The variation of the polymerization degree is more than $0.21ap$ (eq12, Supplementary Information), at least 7 times as much as the cross-scanning contribution. It is likely that the 26 nm size is generated from the laser fluctuation, rather than the crossing structure.

Our Reply : We thank the reviewer for the critical comments. After careful evaluation, we think that the simulation is inadequate to explain the feature size of the HSQ features by FsLDW. Thus, we delete the model and description in the revised manuscript to avoid misunderstanding. Nevertheless, the 26 nm feature size is achieved through the cross-scanning method, instead of the laser fluctuation, as shown in Figure Response 1.

3. Regarding Table S1, how many samples did the authors count? A broad distribution of the hole/line sizes can be observed through Figure 6. In this case, the distribution is not negligible in the simulation of the feature size (X1 and X2 in eq 7, Supplementary Information).

Our Reply: We thank the reviewer for the comments. Regarding Table S1, we counted one sample to investigate detailedly the change of the hole/line sizes and height of the 3D HSQ microstructure under thermal treatment. In the simulation,

we use the feature sizes in Fig. 2 in the previous manuscript (Supplementary Fig. 3 in the revised manuscript), instead of the data in Table S1 and Fig. 6 (Table S1 and Fig. 5 in the revised manuscript). Thus, we think the distribution of the hole/line sizes would not affect the simulation of the feature size in equation 7. Besides, we delete the simulation in the revised manuscript since it is inadequate to explain the feature size of the HSQ features by FsLDW.

4. Regarding Figure 3e, slower scanning speed means higher energy input to a fixed area on the HSQ film, but why does the slower speed lead to a higher damage threshold?

Our Reply: We thank the reviewer for the comment. We agree with the reviewer that slower scanning speed means higher energy input to a fixed area on the HSQ film, and would result in lower damage threshold. In Fig. 3e in the previous manuscript, the damage threshold for the scanning speed of 5 $\mu\text{m/s}$ (2.42 TW/cm^2) is slightly higher than that of the scanning speed of 10, 15 and 20 $\mu\text{m/s}$ (2.33 TW/cm^2 , the damage threshold for the scanning speed of 10, 15 and 20 $\mu\text{m/s}$ is almost identical). The difference of the damage threshold for the scanning speed of 5 $\mu\text{m/s}$ and 10/15/20 $\mu\text{m/s}$ is very small, about 3.8%. The difference may come from several reasons, including laser fluctuation, the deviation of the focus relative to the substrate. *Firstly*, the fluctuation of the fs laser intensity is about 1%, resulting in the difference of the laser damage threshold. *Secondly*, the focus condition of the laser beam would vary slightly during the FsLDW of HSQ. The deviation of the focus relative to the substrate would also result in little change in the damage threshold. In the previous manuscript, we tried to find the fabrication window, in which stable polymerization occurs for HSQ by FsLDW. Much more work is needed to depict the phenomenon near the damage of HSQ by FsLDW, and explore the mechanism of the damage of HSQ. Nevertheless, the main topic of the manuscript is the high-resolution fabrication of HSQ by NIR multi-photon lithography, and the potential applications of the 3D HSQ features. The study of the damage of HSQ by FsLDW is beyond the main topic of the manuscript. To avoid confusion and incomplete demonstration, we delete the Fig. 3 and the discussion in the revised manuscript. The damage of the HSQ by the fs laser irradiation will be systematically studied and discussed in our future work.

5. Supplementary Fig. 4f (Line 115, Page 6) was missed.

Our Reply: We thank the reviewer for the constructive comment. We have corrected the mistake in the revised manuscript by delete “Supplementary Fig. 4f”.

REVIEWER COMMENTS

Reviewer #1 (Remarks to the Author):

The Authors have taken into account most of the amendments satisfactory.

Left remarks:

- line 141: "...freestanding microstructures, which are directly constructed by FSLDW on the substrate." - since they are on attached directly to the substrate, I would not call them freestanding, but rather like "freelying" or somehow as freestanding sounds like standing erect, not lying on.
- 4.2: the NA of the used objective is not mentioned within the Methods, just in the supplementary, though it is an important detail and could be included.

Reviewer #2 (Remarks to the Author):

To editors.

Minor:

1.) The authors claim in the answer to rev. 1 : 'We believe that shrinkage is not the dominant formation mechanism for the 26 nm features, although it may occur during the development process.'

Hence as an answer to rev.2 comments authors write - To "Magnified SEM image indicates that the linewidth of the upper layer grid is smaller than that of the bottom layer, which is probably due to the shrinkage of the nanowires in the development process and surface proximity effects induced by the non-crosslinked HSQ oligomer". - is a bit contradictory. I believe some additional statistical information regarding the changes in feature size between the upper -/lower- layer would be interesting.

2.) The tip deconvolution of the AFM images, will show even better results on feature sizes (comparable or even better than the SEM images). I recommend to perform it.

Reviewer #3 (Remarks to the Author):

Thanks for the response. However, the concerns raised has not been clearly and fully addressed. Some of the author's claims I questioned were just deleted, while no persuasive arguments were added.

Regarding the innovation issue (my first comment), the results demonstrating the 3D fabrication technology advantages were still missed. Millimeter-sized HSQ structures, rather than the micro-sized ones need to be produced by the proposed fs laser pathway, because the large-scale fabrication is a crucial characteristic compared with E-beam. The fabrication speed is also required to be measured/calculated.

As mentioned by the first reviewer, the 26 nm feature was shown in a very restricted way. Shown in Figure Response 1, the width of major parts of the HSQ lines is around 100 nm, the 26 nm feature actually is a "by-product" during the fabrication. A width distribution graph is needed for the reviewer to analysis the practical use of the discovery, as suggested in my third comment.

Regarding the crosslink fabrication, there is no simulation model of the HSQ feature in the revised manuscript. The relationship between the feature and the line spacing needs to be explored experimentally. According to the author's discussion, it can be inferred that larger the difference between the X-axis spacing and the Y-axis one, less the HSQ feature. Why were they always same in the laser fabrication?

Another fatal problem is: since the authors failed to provide direct evidences of the multi-photon lithography (a similar question was raised by the second reviewer), why did they claim the concept in the title?

In summary, although what the authors found was quite notable, they have not shown a clear advantage of the FSDLW and provided a persuasive mechanism behind the discovery, so I do not recommend to accept the paper for publication in Nature Communications.

Point-by-point Response to the Reviewers' Comments

We thank the reviewers for their careful reading of our manuscript and valuable comments. Their constructive comments and suggestions have greatly helped us improve the quality of our manuscript. The following is our point-by-point response to the reviewers' comments.

REVIEWER COMMENTS

Reviewer #1 (Remarks to the Author):

The Authors have taken into account most of the amendments satisfactory.

Our Reply: We thank the reviewer for the positive assessments and valuable comments, which have greatly helped us to improve the quality of the manuscript.

Left remarks:

- line 141: "...freestanding microstructures, which are directly constructed by FsLDW on the substrate." - since they are on attached directly to the substrate, I would not call them freestanding, but rather like "freelying" or somehow as freestanding sounds like standing erect, not lying on.

Our Reply: We thank the reviewer for the constructive comment. We agree with the reviewer that the HSQ microstructure is lying on the substrate. We change the description in the revised manuscript (Line 141-Line 143). "It's worth noting that 33 nm and 26 nm HSQ features are both freelying microstructures, which are directly constructed by FsLDW on the substrate."

- 4.2: the NA of the used objective is not mentioned within the Methods, just in the supplementary, though it is an important detail and could be included.

Our Reply: We thank the reviewer for the constructive comment. We add the description of the NA of the used objective in the revised manuscript (Line 392-Line 394). " The optical setup is shown in Supplementary Fig. 1a, in which the 780 nm fs laser beam is focused by an oil-immersion objective lens (N.A.=1.45, 100 ×, Olympus)."

Reviewer #2 (Remarks to the Author):

To editors.

Minor:

1.) The authors claim in the answer to rev. 1: 'We believe that shrinkage is not the dominant formation mechanism for the 26 nm features, although it may occur during the development

process.'

Hence as an answer to rev.2 comments authors write - To "Magnified SEM image indicates that the linewidth of the upper layer grid is smaller than that of the bottom layer, which is probably due to the shrinkage of the nanowires in the development process and surface proximity effects induced by the non-crosslinked HSQ oligomer". - is a bit contradictory. I believe some additional statistical information regarding the changes in feature size between the upper -/lower- layer would be interesting.

Our Reply: We thank the reviewer for the thoughtful comment. In the FsLDW, shrinkage would occur due to the inadequate crosslinking of the photoresist (the removal of the unpolymerized monomers and oligomers) during the development process. For the HSQ features lying on the substrate, the shrinkage is negligible due to the limitation of the substrate and the surface proximity effects, which effectively suppress the removal of the oligomers. However, the shrinkage of suspended features is non-negligible, because the suspended features are surrounded by the developer, which will remove the unpolymerized monomers and oligomers, leading to the non-negligible shrinkage. As a result, the size of the upper layer HSQ features (suspended features) is smaller than that of the lower layer HSQ features (lying on the substrate) in Figure 4, which is mainly attributed to the shrinkage effect and the surface proximity effects.

Figure Response 1. a 45° tilted-view SEM image of the 3D double layer HSQ microstructure. b The axial size of the upper layer and bottom layer HSQ features

We agree with the reviewer that statistical information regarding the changes in feature size between the upper -/lower- layer would be interesting. We analyzed the axial size of the HSQ features for the upper layer and bottom layer in Figure 4g, as shown in Figure Response 1. The upper layer HSQ features are suspended microstructures for Line 1-5. The bottom layer HSQ features are freely lying on the substrate for Line 2-5, while the bottom layer for Line 1 is suspended microstructures. For Line 2-5, the axial size of the bottom layer HSQ features is around 200 nm, while the axial size of the upper layer HSQ features is around 100 nm. The size of the bottom layer HSQ features is about two times larger than that of the upper layer HSQ features, which is mainly attributed to the shrinkage of the HSQ features during the development process and surface proximity effects induced by the non-crosslinked HSQ oligomers. The axial size of the HSQ features for Line 1 further verifies the above analysis. For line 1, the axial size of the upper layer HSQ features is

around 70 nm, while the axial size of the bottom layer HSQ features is around 100 nm. The axial size of the bottom layer for Line 1 is similar to that of the upper-layer for Line 2-5, while much smaller than that of the bottom layer of Line 2-5. The difference is mainly attributed to the fact that the bottom layer of Line 1 is suspended microstructures, instead of freelying microstructures. Compared to the suspended HSQ microstructures, the freelying HSQ microstructures are larger due to the suppressed shrinkage during the development process and the surface proximity effects induced by the noncrosslinked HSQ oligomers.

Accordingly, we plot the statistical information regarding the changes in feature size between the upper -/lower- layer in the Supplementary Fig. 8, and added the discussion in the revised supporting information. “We analyzed the axial size of the HSQ features for the upper layer and bottom layer in Figure 4g, as shown in Supplementary Figure 8. The upper layer HSQ features are suspended microstrucutres for Line 1-5. The bottom layer HSQ features are freelying on the substrate for Line 2-5, while the bottom layer for Line 1 is suspended microstructures. For Line 2-5, the axial size of the bottom layer HSQ features is around 200 nm, while the axial size of the upper layer HSQ features is around 100 nm. The size of the bottom layer is about two times larger than that of the upper layer, which is mainly attributed to the shrinkage of the nanowires during the development process and surface proximity effects induced by the non-crosslinked HSQ oligomers. The axial size of the HSQ features for Line 1 further verifies the above analysis. For line 1, the axial size of the upper layer HSQ features is around 70 nm, while the axial size of the bottom layer HSQ features is around 100 nm. The axial size of the bottom layer for Line 1 is similar to that of the upper layer for Line 2-5, while much smaller than that of the bottom layer of Line 2-5. The difference is mainly attributed to the fact that the bottom layer of Line 1 is suspended microstructures, instead of freelying microstructures. Compared to the suspended HSQ microstructures, the freelying HSQ microstructures are larger due to the suppressed shrinkage during the development process and the surface proximity effects induced by the noncrosslinked HSQ oligomers.”

2.) The tip deconvolution of the AFM images, will show even better results on feature sizes (comparable or even better than the SEM images). I recommend to perform it.

Our Reply : We thank the reviewer for the constructive comment. According to the reviewer’s suggestion, we have performed the tip deconvolution of the AFM images by employing the Matlab 2020a and ImageJ software, as shown in Figure Response 2. The height and full width at half maximum (FWHM) of the narrowest HSQ nanowire is 26 nm and 36 nm, which are smaller than the results shown in the Supplementary Figure 3. The FWHM of the narrowest HSQ nanowire before and after tip deconvolution are 39 and 36 nm, which are slightly larger than the result in the SEM image (Figure 2). The slight deviation of the AFM image and SEM image is mainly attributed to the impact of the tip during AFM measurement. Although the height is strictly accurate for AFM measurement, the width is inclined to deviate from the real value due to the influence of the tip, especially for the features with very small feature size. As a result, the feature size of the HSQ features is evaluated by using SEM images in the revised manuscript.

Figure Response 2. **a** AFM image of the HSQ nanowires fabricated by FsLDW with different scanning speed from 5 to 100 $\mu\text{m/s}$. **c** AFM image of the HSQ nanowire fabricated by FsLDW with scanning speed of 100 $\mu\text{m/s}$. **d** Z-profile of the HSQ nanowires shown in **c**

Reviewer #3 (Remarks to the Author):

Thanks for the response. However, the concerns raised has not been clearly and fully addressed. Some of the author's claims I questioned were just deleted, while no persuasive arguments were added.

Our Reply: We thank the reviewer for the careful reading of the manuscript and thoughtful comments. The comments have greatly improved the quality of the revised manuscript. We deleted some content because the model and description is inadequate to explain the formation mechanism of the HSQ features. According to the reviewer's comments, we added some experimental data in the revised manuscript to further investigate the formation mechanism of the nanoscale HSQ features. We hope the revised manuscript would adequately address the reviewer's concerns.

1.) Regarding the innovation issue (my first comment), the results demonstrating the 3D

fabrication technology advantages were still missed. Millimeter-sized HSQ structures, rather than the micro-sized ones need to be produced by the proposed fs laser pathway, because the large-scale fabrication is a crucial characteristic compared with E-beam. The fabrication speed is also required to be measured/calculated.

Our Reply: We thank the reviewer for the critical comment. The advantages of femtosecond laser direct writing (FsLDW) of HSQ are clear. *Firstly*, the feature size of the HSQ nanowire by FsLDW is as high as 33 and 26 nm (Fig. 2b), which is only 1/24 and 1/30 of the 780 nm fs laser irradiation. *Secondly*, 3D HSQ microstructures with thermal and chemical resistance have been obtained by FsLDW (Fig. 4 and Fig. 5). Especially, 3D HSQ microstructure with structural colour has been successfully achieved by FsLDW. *Thirdly*, FsLDW is performed in ambient atmosphere, and on insulating substrate, such as glass. However, E-Beam lithography is performed in vacuum atmosphere and on conducting substrates. Conductive films are needed to overcome the charge accumulation on the insulating substrates^{1,2}, which would significantly degrade the quality of the microstructures, or even lead to the failure of patterning. The vacuum atmosphere and the conductive films will dramatically add the complexity of the E-beam lithography.

Figure Response 3. **a** Photo of 1 mm × 0.2 mm HSQ grating structure by FsLDW. **b** SEM image of the HSQ grating structure with the periodicity of 4 μm. **c** The diffraction pattern observed in transmission for the HSQ grating structure

According to the reviewer's comment, we have fabricated millimeter-sized HSQ structure by FsLDW. As shown in Figure Response 3a, HSQ grating structure with 1 mm × 0.2 mm has been successfully constructed. SEM image reveals no defect in the uniform HSQ grating structure, in which the periodicity is 4 μm (Figure Response 3b). Figure Response 3c shows the diffraction pattern observed in transmission for the HSQ grating structures. The diffracted laser beam spots up to 7th order can be well recognized, indicating the high quality of the HSQ grating structure fabricated by FsLDW.

The fabrication speed of the millimeter-sized HSQ structure by FsLDW is 100 μm/s, which is the commonly used scanning speed in FsLDW. Nevertheless, S. K. Saha recently demonstrated femtosecond projection two-photon polymerization (FP-TPL) technique to fabricate arbitrary 3D complex structures with submicrometer resolution and high-throughput³. We think that the fabrication speed of HSQ could be dramatically increased by using the above parallelizing femtosecond laser direct writing lithography. We will perform the high-throughput fabrication of 3D HSQ microstructures with high resolution and high speed in our future work.

References

- [1] Liu, J. K., Li, Q. Q., Ren, M. X., Zhang, L. H., Chen, M. & Fan, S. S. Graphene as discharge layer for electron beam lithography on insulating substrate. *Appl. Phys. Lett.* 103, 113107 (2013).
- [2] Dobisz, E. A., Bass, R., Brandow, S. L., Chen, M. S. & Dressick, W. J. Electroless metal discharge layers for electron beam lithography. *Appl. Phys. Lett.* 82(3), 478 (2003).
- [3] Saha, S. K., Wang, D., Nguyen, V. H., Chang, Y., Oakdale, J. S. & Chen, S.-C. Scalable submicrometer additive manufacturing. *Science* 366, 105–109 (2019).

2.) As mentioned by the first reviewer, the 26 nm feature was shown in a very restricted way. Shown in Figure Response 1, the width of major parts of the HSQ lines is around 100 nm, the 26 nm feature actually is a “by-product” during the fabrication. A width distribution graph is needed for the reviewer to analysis the practical use of the discovery, as suggested in my third comment.

Our Reply: We thank the reviewer for the comment. We agree with the reviewer that the 26 nm HSQ feature was shown in a restricted way. In our manuscript, the 26 nm is achieved by employing cross-scanning method through FsLDW. Nevertheless, we think the 26 nm feature is not a “by-product” during the fabrication. Indeed, the 26 nm feature is reproducible and of practical use. To depict the reproducibility of the HSQ feature with nanoscale size, we fabricate the HSQ features by cross-scanning method, as shown in Figure Response 4a. The feature size of the HSQ (the smallest size of the HSQ features) is plotted in Figure Response 4c, in which the HSQ feature size changes from 26 nm to 60 nm. The results in Supplementary Figure 5 and Figure Response 4 depict that 26 nm HSQ feature is reproducible under optimized conditions.

Figure Response 4. a SEM image of HSQ features fabricated by FsLDW using cross-scanning

method with the spacing of 0.5 μm . **b** Magnified SEM image of 26 nm HSQ feature. **c** The feature size of the HSQ features fabricated by FsLDW using cross-scanning method. **d** The size distribution graph of the 26 nm HSQ feature

Furthermore, the width distribution graph of the 26 nm HSQ feature is plotted in Figure Response 4d, in which the size of the HSQ feature varies from 26 nm to 104 nm. As discussed in the revised manuscript, the 26 nm is attributed to the interference between the adjacent lines, resulting in the further approaching of the laser intensity threshold. The widening of the HSQ feature is mainly attributed to the surface proximity effects induced by the non-crosslinked HSQ oligomers. Although 26 nm HSQ feature is obtained in the cross-scanning method, it would find practical use in the construction of functional microstructures, such as SERS substrate for Raman enhancement, FinFET, and so on.

3.) Regarding the crosslink fabrication, there is no simulation model of the HSQ feature in the revised manuscript. The relationship between the feature and the line spacing needs to be explored experimentally. According to the author's discussion, it can be inferred that larger the difference between the X-axis spacing and the Y-axis one, less the HSQ feature. Why were they always same in the laser fabrication?

Our Reply: We thank the reviewer for the constructive comment. We agree with the reviewer that the relationship between the feature and the line spacing needs to be explored experimentally. We studied the relationship between the feature and the line spacing of 0.5, 1, and 2 μm fabricated by FsLDW with single-scanning method and cross-scanning method, as shown in Figure Response 5.

For single-scanning method, the narrowest linewidth of the HSQ nanowire by FsLDW is 53, 58 and 72 nm for the line spacing of 0.5, 1, and 2, respectively (Figure Response 5a, c, and e). The threshold of laser intensity is 1.56 TW/cm^2 for the line spacing of 1 and 2 μm , which is larger than that of the line spacing of 0.5 μm , i.e. 1.45 TW/cm^2 (Figure Response 5d). The decrease of the threshold of 0.5 μm spacing is attributed to the interference between adjacent lines. Moreover, the interference between adjacent lines also lead to the bigger linewidth of HSQ for the 0.5 μm line spacing. For cross-scanning method, the smallest feature size of HSQ features is 26, 71 and 77 nm for the line spacing of 0.5, 1, and 2 μm , respectively (Figure Response 5b, d, and f). Compared to the single-scanning method, the feature size of the HSQ dramatically decreases from 53 to 26 nm, and the threshold is further decreased to 1.42 TW/cm^2 (Figure Response 5h). The decrease of the feature size and threshold is ascribed to the interference between cross and adjacent lines.

We systematically investigated the dependence of the HSQ feature size on the line spacing, it can be inferred that smaller the line spacing, less the HSQ feature size, which is attributed to the interference between cross and adjacent lines.

Figure Response 5. Feature sizes of HSQ features fabricated by FsLDW using single-scanning method and cross-scanning method with different spacing. SEM images of HSQ nanowires fabricated by FsLDW using single-scanning method with the spacing of **a** 2, **c** 1, and **e** 0.5 μm, respectively. SEM images of HSQ features fabricated by FsLDW using cross-scanning method with the spacing of **b** 2, **d** 1, and **f** 0.5 μm, respectively. **g** The influence of laser intensity and line spacing on the feature size of HSQ features fabricated by FsLDW using single-scanning method. **h** The influence of laser intensity and line spacing on the feature size of HSQ features fabricated by FsLDW using cross-scanning method

4.) Another fatal problem is: since the authors failed to provide direct evidences of the

multi-photon lithography (a similar question was raised by the second reviewer), why did they claim the concept in the title?

Our Reply: We thank the reviewer for the comment. The construction of HSQ features by FsLDW is ascribed to multi-photon lithography due to two reasons. *Firstly*, HSQ does not absorb light in the wavelength range between 200 and 800 nm (Supplementary Fig. 2), but exhibits photocuring sensitivity at 157 nm¹, which suggest that HSQ can't be cured through single photon process by the irradiation of 780 nm femtosecond laser. It's feasible that the fabrication of HSQ features by FsLDW is attributed to multi-photon lithography. *Secondly*, 3D HSQ microstructures have been successfully fabricated by the FsLDW of HSQ (Figure 4, 5, 6 and Supplementary Figure 7). The capability of constructing 3D micro-structures is the intrinsic characteristic of multi-photon lithography. As a result, we depict that the FsLDW of HSQ is attributed to multi-photon lithography. Nevertheless, we will continue to investigate the direct evidences of the multi-photon absorbance process of HSQ by FsLDW in our future work.

References

[1] Peuker, M., Lim, M. H., Smith, H. J., Morton, R., van Langen-Suurling, A. K., Romijn, J., van der Drift, EWJM & van Delft, FCMJM Hydrogen silsesquioxane, a high-resolution negative tone e-beam resist, investigated for its applicability in photon-based lithographies. *Microelectron. Eng.* 61-62, 803-809 (2002).

5.) In summary, although what the authors found was quite notable, they have not shown a clear advantage of the FSDLW and provided a persuasive mechanism behind the discovery, so I do not recommend to accept the paper for publication in Nature Communications.

Our Reply: We thank the reviewer for the critical comment. The advantages of the FsLDW of HSQ is clear. Firstly, the feature size of 26 nm, $\lambda/30$, can be achieved by FsLDW. Secondly, 3D HSQ features with thermal and chemical resistance have been obtained by FsLDW. Thirdly, FsLDW of HSQ features is performed in ambient condition, instead of strict operation atmosphere, for example, vacuum condition. Meanwhile, we added experimental data to further verify the mechanism behind the discovery. Accordingly, we have revised the manuscript according to the reviewer's valuable comments and suggestions. We hope the revised manuscript will be suitable for publication in Nature Communications.

REVIEWER COMMENTS

Reviewer #2 (Remarks to the Author):

The authors have taken into account all revisions. No additional revision is required.

Reviewer #3 (Remarks to the Author):

Thanks for the author's great effort to improve the manuscript quality. Unfortunately, I still do not recommend to accept the paper for publication in Nature Communications, because 1) the laser fabrication of 26 nm HSQ lines is not verified to be controllable, and 2) no direct evidence can support the multi-photon lithography.

In Figure Response 4b, actually there are at least 2 HSQ lines fabricated by FSLDW using the cross-scanning method under same parameters. The bottom one seems to be discontinuous, and the top one shows a narrowest width of 26 nm. These broken lines has not been taken into account in Figure Response 5g-h. The cross-scanning fabrication induces a shrinkage effect to the HSQ lines, so some of them are broken, some widths decrease to 40-60 nm. Only a slight percentage of the widths are around 26 nm. If more SEM images with higher magnifications are taken, it is possible to observed a line narrower than 26 nm from a large amount of them. Consequently, the fabrication method cannot be verified to be controllable, because the 26 nm lines are destined to be accompanied with lots of thick and broken ones. A pathway with laser fabrication parameters, realizing that a major part of HSQ lines demonstrate widths of 26 nm, should be provided, and a real application can only be generated from a controllable lithography.

The authors' claim that HSQ does not absorb light in the wavelength range between 200 and 800 nm just excludes the possibility of single-photon absorption from HSQ, and the multi-photon absorption is not the only way to get 3D structures. It is necessary to show some direct evidences such as Z-scan data.

Point-by-point Response to the Reviewers' Comments

We thank the reviewers for their careful reading of our manuscript and valuable comments. Their constructive comments and suggestions have greatly helped us improve the quality of our manuscript. The following summaries are our point-by-point response to the reviewers' comments.

REVIEWER COMMENTS

Reviewer #2 (Remarks to the Author):

The authors have taken into account all revisions. No additional revision is required.

Our Reply: We thank the reviewer for the positive assessments and constructive comments.

Reviewer #3 (Remarks to the Author):

Thanks for the author's great effort to improve the manuscript quality. Unfortunately, I still do not recommend to accept the paper for publication in Nature Communications, because 1) the laser fabrication of 26 nm HSQ lines is not verified to be controllable, and 2) no direct evidence can support the multi-photon lithography.

Our Reply: We thank the reviewer for the careful reading of the manuscript and helpful comments. The comments have greatly improved the quality of the revised manuscript. We have demonstrated the feasibility to fabricate 26 nm HSQ features by four laser fabrication strategies using FsLDW, indicating the reproducibility of fabricating 26 nm HSQ features by FsLDW. Meanwhile, we have also discussed the fabrication parameters needed to be optimized for the controllable fabrication of 26 nm HSQ features. Moreover, we have achieved preliminary experimental evidence to support the multi-photon lithography for the FsLDW of HSQ. We hope the revised manuscript would adequately address the reviewer's concerns.

Comment 1:

In Figure Response 4b, actually there are at least 2 HSQ lines fabricated by FsLDW using the cross-scanning method under same parameters. The bottom one seems to be discontinuous, and the top one shows a narrowest width of 26 nm. These broken lines has not been taken into

account in Figure Response 5g-h. The cross-scanning fabrication induces a shrinkage effect to the HSQ lines, so some of them are broken, some widths decrease to 40-60 nm. Only a slight percentage of the widths are around 26 nm. If more SEM images with higher magnifications are taken, it is possible to observe a line narrower than 26 nm from a large amount of them. Consequently, the fabrication method cannot be verified to be controllable, because the 26 nm lines are destined to be accompanied with lots of thick and broken ones. A pathway with laser fabrication parameters, realizing that a major part of HSQ lines demonstrate widths of 26 nm, should be provided, and a real application can only be generated from a controllable lithography.

Our Reply: We thank the reviewer for the helpful comment. FsLDW of 26 nm HSQ feature size is reproducible, which has been verified by previous experimental results. In principle, it's feasible that nanoscale HSQ features with feature size of 26 nm can be fabricated by FsLDW. Nevertheless, it's still difficult to fabricate 26nm HSQ features from a controllable lithography since the size of the nanoscale HSQ features could be greatly influenced by the fabrication parameters, such as laser intensity fluctuation, focusing condition, roughness of the glass substrate, post-polymerization shrinkage, and working environments. It's reasonable that some discontinuous and broken HSQ features are obtained in our current fabrication setup when the size is approaching 26 nm. In order to further depict the reproducibility of 26 nm HSQ features by FsLDW, we performed the FsLDW of HSQ features by employing four laser fabrication strategies, as shown in Figure Response 1-4. We have detailedly investigated the influence of laser intensity and scanning speed on the feature size of the HSQ features by FsLDW using cross-scanning method as follows.

(i) As shown in Figure Response 1, we changed the laser intensity in one direction (Line X), while kept the laser intensity in the perpendicular direction (Line Y). In the fabrication, the laser scanning speed is kept at 10 $\mu\text{m/s}$. When the laser intensity is gradually decreased to 1.42 TW/cm^2 , HSQ of 26 nm feature size is obtained by FsLDW.

(ii) We simultaneously decreased the laser intensity in two perpendicular directions (both Line X and Line Y), while kept the laser scanning speed of 10 $\mu\text{m/s}$. 25 nm HSQ feature is successfully achieved when the laser intensity is decreased to 1.46 TW/cm^2 , as shown in Figure Response 2.

(iii) As shown in Figure Response 3, we changed the laser scanning speed in one direction (Line X), while kept the laser scanning speed in the perpendicular direction (Line Y). In the fabrication, the laser intensity is kept at 1.54 TW/cm^2 . With the increase of laser scanning speed from $10 \text{ }\mu\text{m/s}$ to $85 \text{ }\mu\text{m/s}$, the feature size of HSQ features gradually decreases from 257 nm to 22 nm .

(iv) We simultaneously changed the laser scanning speed in two perpendicular directions (both Line X and Line Y). In the fabrication, the laser intensity is kept at 1.68 TW/cm^2 . With the increase of laser scanning speed from $10 \text{ }\mu\text{m/s}$ to $105 \text{ }\mu\text{m/s}$, the feature size of HSQ features gradually decreases from 297 nm to 25 nm .

Figure Response 1. Feature sizes of HSQ features fabricated by FsLDW using cross-scanning method with different laser intensities in one direction. **a** The illustration of HSQ features fabricated by FsLDW using cross-scanning method with different laser intensities in one direction. The scanning speed is kept at $10 \text{ }\mu\text{m/s}$. **b** SEM image of HSQ features fabricated by FsLDW using cross-scanning method with the spacing of $0.5 \text{ }\mu\text{m}$. **c** The influence of laser intensity on the feature size of HSQ features fabricated by FsLDW marked by yellow dot line in **b**. **d** SEM image of HSQ feature with feature size of 26 nm marked by red line in **b**.

Figure Response 2. Feature sizes of HSQ features fabricated by FsLDW using cross-scanning method with different laser intensities in two perpendicular directions. The scanning speed is kept at $10 \mu\text{m/s}$. **a** The illustration of HSQ features fabricated by FsLDW using cross-scanning method with different laser intensities in two orthogonal directions. **b** SEM image of HSQ features fabricated by FsLDW using cross-scanning method with different laser intensities in two orthogonal directions. **c** The influence of laser intensity on the feature size of HSQ features fabricated by FsLDW marked by yellow dot line in **b**. **d** SEM image of HSQ feature with feature size of 25 nm marked by red line in **b**.

Figure Response 3. Feature sizes of HSQ features fabricated by FsLDW using cross-scanning method with different scanning speed in one direction. The laser intensity is kept at 1.54 TW/cm^2 . **a** The illustration of HSQ features fabricated by FsLDW using cross-scanning method with different scanning speed in one direction. **b** SEM image of HSQ features fabricated by FsLDW using cross-scanning method with different scanning speed in one direction. **c** The influence of scanning speed on the feature size of HSQ features fabricated by FsLDW marked by yellow dot line in **b**. **d** SEM image of HSQ feature with feature size of 22 nm marked by red line in **b**.

Figure Response 4. Feature sizes of HSQ features fabricated by FsLDW using cross-scanning method with different scanning speed in two perpendicular directions. The laser intensity is kept at 1.68 TW/cm^2 . **a** The illustration of HSQ features fabricated by FsLDW using cross-scanning method with different scanning speed in two orthogonal directions. **b** SEM image of HSQ features fabricated by FsLDW using cross-scanning method with different scanning speed in two orthogonal directions. **c** The influence of scanning speed on the feature size of HSQ features fabricated by FsLDW marked by yellow dot line in **b**. **d** SEM image of HSQ feature with feature size of 25 nm marked by red line in **b**.

Herein, 26 nm feature size is successfully achieved by precisely tuning laser intensity or laser scanning speed employing cross-scanning method. The fabrication of 26 nm feature size of HSQ features by FsLDW is reproducible under optimized conditions. However, it's still difficult to construct 26 nm HSQ features by FsLDW at will. The reasons mainly lie in the following aspects.

- (i) The fluctuation of laser intensity of 780 nm femtosecond laser. According to the manual

of the femtosecond laser, the fluctuation of the laser intensity is $< \pm 1\%$. Nevertheless, the femtosecond laser employed in this study has been manufactured for more than 16 years (Tsunami, Spectra Physics, October 2005), which probably causes higher fluctuation of laser intensity. The influence of the fluctuation of the laser intensity is non-negligible, especially when the laser intensity is approaching to the threshold due to the nonlinear absorption of HSQ. Prof. Yang has reported that the yield dropped from 100% at 1.07 times of measured minimum power to 5% at the minimum power (for suspended 7 nm feature in a commercial organic photoresist) due to the strong sensitivity of the feature formation to laser dose¹. As a result, the fluctuation of laser intensity has hindered the fabrication of uniform nanoscale HSQ features.

(ii) It's difficult to produce an identical focusing condition for different HSQ features. As we all know, the focusing condition is critically important for the feature size in the FsLDW due to nonlinear absorption of HSQ for 780 nm femtosecond laser beam. We do not build an automatic focalizing setup on the FsLDW system to precisely control the focusing condition, resulting in the difficulty in making an identical focusing condition in the fabrication of HSQ features. Even for the fabrication of HSQ features in a small region, the deviation of focusing condition is also non-negligible owing to the surface roughness of the glass substrate, the tilt of the 3D piezostage and/or the stage of the optical microscopy, and so on.

(iii) The influence of thickness nonuniformity of the HSQ films. Previous report has demonstrated the importance of the film thickness on the feature size of HSQ features by EBM lithography. In EBM lithography, sub-10 nm HSQ features can be fabricated by using thinner HSQ films (about 10 nm thickness) obtained by spin coating of diluted HSQ solution². Herein, the roughness of the glass substrate would produce HSQ film with thickness nonuniformity during spin coating. As a result, smaller HSQ features could be fabricated in the local site with thinner thickness.

(iv) The post-polymerization shrinkage. The post-polymerization shrinkage of features is a common phenomenon in femtosecond laser direct writing, especially for the suspended features. The shrinkage often occurs in the developing process, owing to the removal of the unpolymerized monomers and/or oligomers. The post-polymerization shrinkage facilitated the formation of suspended nanoscale features.^{3, 4} However, the understanding and manipulation of the post-polymerization shrinkage is preliminary since the shrinkage process is influenced

by complex physical and chemical process. Post-polymerization shrinkage may be a possible reason, but not the dominant reason for the fabrication of nanoscale HSQ features, since the shrinkage is restricted by the substrate for the free-lying features.

(v) The influence of the working environments. HSQ is extremely sensitive to the atmosphere conditions, such as humidity and/or temperature. Moreover, the building vibration will also affect the fabrication process of the nanoscale HSQ features. The working environment would affect the uniformity of the HSQ feature size.

In the manuscript, we demonstrated the fabrication reproducibility of 26 nm HSQ features by FsLDW. In principle, it's feasible that 26 nm HSQ features can be fabricated by FsLDW. Meanwhile, we are aware that the size of the nanoscale HSQ features could be greatly influenced by the fabrication parameters, such as laser intensity fluctuation, focusing condition, roughness of the glass substrate, post-polymerization shrinkage, and working environments. We agree with the reviewer that it is important to construct 26 nm HSQ features in a controllable method. We believe the controllable lithography of 26 nm HSQ features can be achieved by optimizing the above fabrication parameters.

References

- [1] Wang, S., Yu, Y., Liu, H., Lim, K. T P, Srinivasan, B. M., Zhang Y. W.& Yang KW J. Sub-10-nm suspended nano-web formation by direct laser writing. *Nano Futures* 2, 025006 (2018).
- [2] Yang, KW J., Cord, B., Duan, H., Berggren, K. K., Klingfus, J., Nam S.-W., Kim, K.-B. & Rooks, M. J. Understanding of hydrogen silsesquioxane electron resist for sub-5-nm-half-pitch lithography, *J. Vac. Sci. Technol. B* 27(6), 2622-2627 (2009).
- [3] Takada, K., Wu, D., Chen, Q.-D., Shoji, S., Xia, H., Kawata, S. & Sun H.-B. Size-dependent behaviors of femtosecond laser-prototyped polymer micronanowires. *Opt. Lett.* 34(5), 566-568 (2009).
- [4] Tan, D. F., Li, Y., Qi, F. J., Yang, H., Gong, Q. H., Dong X. Z. & Duan X.-M. Reduction in feature size of two-photon polymerization using SCR500. *Appl. Phys. Lett.* 90, 071106 (2007).

Comment 2:

The authors' claim that HSQ does not absorb light in the wavelength range between 200 and 800 nm just excludes the possibility of single-photon absorption from HSQ, and the multi-photon absorption is not the only way to get 3D structures. It is necessary to show some direct evidences such as Z-scan data.

Our Reply: We appreciate the reviewer for the insightful comment. The construction of HSQ features by FsLDW is ascribed to multi-photon lithography due to the fact that HSQ does not absorb light in the wavelength range between 200 and 800 nm (Supplementary Fig. 2), but exhibits photocuring sensitivity at 157 nm¹. The phenomenon suggests that HSQ can't be cured through single-photon process by the irradiation of 780 nm femtosecond laser. It's feasible that the fabrication of HSQ features by FsLDW is attributed to multi-photon lithography. According to the reviewer's suggestion, we have measured the multi-photon absorption characteristic of HSQ solution by using Z-scan method, but failed to obtain effective results. The laser intensity threshold for the multi-photon absorption of the solvent is lower than that of the HSQ molecules. Thus, we did not detect effective multi-photon absorption of HSQ until the appearance of white light of the solvent induced by the 780 nm femtosecond laser.

To achieve the direct evidences, we have also performed in-site measurement of the nonlinear absorption order in HSQ using the method proposed by Prof. Mueller and Prof. Wegener². **The experimental results suggest that multi-photon absorption occurs in HSQ irradiated by the 780 nm femtosecond laser, as shown in Figure Response 5.** Figure Response 5a-j present the SEM images of HSQ line array by FsLDW with the scanning speed from 10 to 100 $\mu\text{m/s}$. For shorter exposure (exposure time smaller than 270 ms, corresponding to a laser scanning speed higher than 50 $\mu\text{m/s}$), the photocuring laser threshold power versus the scanning speed obey the rule²⁻⁴:

$$P_{th} \propto C \times v^{1/N} \quad (1)$$

Where P_{th} is the laser threshold power at a given scanning speed, C is a coefficient associated with the characteristic of the photoresist, v is the scanning speed, and N is the nonlinear absorption order in HSQ. **For HSQ line array by a 780 nm femtosecond laser, N is determined to be 3.83 (Figure Response 5k), indicating multi-photon absorption in HSQ by FsLDW.** As a result, we depict that FsLDW of HSQ is attributed to multi-photon lithography.

Figure Response 5. Photocuring laser threshold power P_{th} as a function of the scanning speed for HSQ by FsLDW. SEM images of HSQ line array by FsLDW with the scanning speed of a 10, b 20, c 30, d 40, e 50, f 60, g 70, h 80, i 90, and j 100 μm/s, respectively. The scale bar is 10 μm. k Experimental data and fitting of the laser threshold power versus the scanning speed in HSQ by FsLDW. The square is the experimental data, and the dash line is the fitting result

Accordingly, we added the description and discussion of the multi-photon lithography in the revised manuscript and supporting information as follows.

(1) In the revised manuscript, we added the description in Page 5, Line112-114, “The multi-photon absorption process is further verified by the in-site measurement of the nonlinear absorption order in HSQ⁴⁶⁻⁴⁸, indicating multi-photon absorption in HSQ by FsLDW (Supplementary Fig. 3).”.

(2) In the revised supporting information, we added Supplementary Figure 3 and the discussion as follows. “We have performed in-site measurement of the nonlinear absorption order in HSQ using the method proposed by Prof. Mueller and Prof. Wegener¹. The experimental results suggest that multi-photon absorption occurs in HSQ irradiated by a 780 nm femtosecond laser, as shown in Supplementary Figure 3. Supplementary Figure 3a-j present the SEM images of HSQ line array by FsLDW with the scanning speed from 10 to 100 $\mu\text{m/s}$. For shorter exposure (exposure time smaller than 270 ms, corresponding to a laser scanning speed higher than 50 $\mu\text{m/s}$), the photocuring laser threshold power versus the scanning speed obey the rule¹⁻³:

$$P_{th} \propto C \times v^{1/N} \quad (1)$$

Where P_{th} is the laser threshold power at a given scanning speed, C is a coefficient associated with the characteristic of the photoresist, v is the scanning speed, and N is the nonlinear absorption order in HSQ. For HSQ line array by a 780 nm femtosecond laser, N is determined to be 3.83 (Supplementary Figure 3k), indicating multi-photon absorption in HSQ by FsLDW. As a result, we depict that FsLDW of HSQ is attributed to multi-photon lithography.”.

References

- [1] Peuker, M., Lim, M. H., Smith, H. J., Morton, R., van Langen-Suurling, A. K., Romijn, J., van der Drift, EWJM. & van Delft, FCMJM. Hydrogen silsesquioxane, a high-resolution negative tone e-beam resist, investigated for its applicability in photon-based lithographies. *Microelectron. Eng.* 61-62, 803-809 (2002).
- [2] Mueller, J. B. Fischer, J., Mayer, F., Kadic, M. & Wegener, M. Polymerization kinetics in three-dimensional direct laser writing. *Adv. Mater.* 26, 6566-6571 (2014).
- [3] Yang, L., Münchinger, A., Kadic, M., Hahn, V., Mayer, F., Blasco, E., Barner-Kowollik, C. & Wegener M. On the schwarzschild Effect in 3D two-photon laser lithography. *Adv. Opt. Mater.* 7, 1901040 (2019).
- [4] Yu, H., Ding, H., Zhang, Q., Gu, Z. & Gu, M. Three-dimensional direct laser writing of PEGda hydrogel microstructures with low threshold power using a green laser beam, *Light: Adv. Manufacturing* 2: 3 (2021).

REVIEWERS' COMMENTS

Reviewer #3 (Remarks to the Author):

Thanks for the author's great effort to improve the manuscript quality. The comments have addressed my second concern regarding the multi-photon HSQ absorption. However, the first concern about the controllable laser fabrication has not been responded to thoroughly. The authors fabricated HSQ networks under different laser intensities and scanning speeds in both X and Y directions. Even though the authors cannot realize large amounts of 26 nm lines, the authors need to demonstrate the possibility of selectivity to get them. The results do not provide a pathway to a high-impact application if the 26 nm one appears randomly. Multiple HSQ networks under the same parameters can be produced, and the width distribution at a fixed location of each network is required to be measured. I also suggest adding some discussions of the pathway to achieve a controllable fabrication in the manuscript.

Point-by-point Response to the Reviewers' Comments

We thank the reviewers for their careful reading of our manuscript and valuable comments. Their constructive comments and suggestions have greatly helped us improve the quality of our manuscript. The following summaries are our point-by-point response to the reviewers' comments.

REVIEWER COMMENTS

Reviewer #3 (Remarks to the Author):

Thanks for the author's great effort to improve the manuscript quality. The comments have addressed my second concern regarding the multi-photon HSQ absorption. However, the first concern about the controllable laser fabrication has not been responded to thoroughly. The authors fabricated HSQ networks under different laser intensities and scanning speeds in both X and Y directions. Even though the authors cannot realize large amounts of 26 nm lines, the authors need to demonstrate the possibility of selectivity to get them. The results do not provide a pathway to a high-impact application if the 26 nm one appears randomly. Multiple HSQ networks under the same parameters can be produced, and the width distribution at a fixed location of each network is required to be measured. I also suggest adding some discussions of the pathway to achieve a controllable fabrication in the manuscript.

Our Reply: We thank the reviewer for the careful reading of the manuscript and constructive comments. The comments have greatly improved the quality of the revised manuscript. We agree with the reviewer that we need to demonstrate the possibility of the selectivity to get large amounts of 26 nm HSQ features.

According to the reviewer's suggestion, we fabricated three HSQ networks with the same parameters, and measured the width distribution at a fixed location of each network, as shown in Figure 1. HSQ networks were fabricated by FsLDW using cross-scanning method with the same laser intensity, while the scanning speed is fixed at 80 $\mu\text{m/s}$ and 20 $\mu\text{m/s}$ for the perpendicular X direction and Y direction. In Figure 1A, we achieved thirteen HSQ features marked by the dashed yellow line, although we designed twenty HSQ features. The size of the HSQ features changes from 21 nm to 86 nm. Three HSQ features have the size of 21, 26 and 28 nm, respectively. In Figure 1B, five HSQ features were obtained, with the size varying from 23 nm to 88 nm. In Figure 1C, six HSQ features were constructed, with the size varying from

Figure 1. A-1, B-1, and C-1 SEM images of HSQ features fabricated by FsLDW using cross-scanning method with the spacing of 0.45 μm. The insets are the magnified SEM images of the HSQ features marked by solid red lines. **A-2, B-2, and C-2** The size distribution of the HSQ features marked by dashed yellow lines in **A-1, B-1, and C-1**.

25 nm to 103 nm. Although we pursue the goal of fabricating large amounts of 26 nm HSQ features, the yield of the 26 nm HSQ features is 15%, 5%, and 5% at a fixed location. The yield is similar to prof. Yang's research on suspended organic features¹, in which the yield dropped from 100% at 1.07 times of measured minimum power to 5% at the minimum power due to the strong sensitivity of the feature formation to laser dose. Meanwhile, HSQ is sensitive to the environmental humidity and storage time, which may also result in the weak uniformity of the nanoscale HSQ features. We believe the large amounts of 26 nm HSQ features could be produced under optimal conditions.

Moreover, we are aware that the size of the nanoscale HSQ features could be greatly

influenced by the experimental parameters, such as laser intensity fluctuation, focusing condition, roughness of the glass substrate, post-polymerization shrinkage, environmental humidity and storage time of HSQ. We think the controllable lithography of 26 nm HSQ features can be achieved by optimizing the experimental parameters.

According to the reviewer's suggestion, we added the discussions in the revised manuscript (Page 7, Line 158-162). **“Meanwhile, the size of the HSQ features could be greatly influenced by the experimental parameters, such as laser intensity fluctuation, focusing condition, roughness of the substrate, post-polymerization shrinkage, environmental humidity and storage time of HSQ. The controllable lithography of 26 nm HSQ features could be achieved by optimizing the experimental parameters.”**

Reference

[1] Wang, S., Yu, Y. Liu, H., Lim, K. T P, Srinivasan, B. M., Zhang Y. W.& Yang KW J. Sub-10-nm suspended nano-web formation by direct laser writing. *Nano Futures* **2**, 025006 (2018).